# Temporal asymmetries in inferring unobserved past and future events

Xinming Xu [1], Ziyan Zhu[2], Xueyao Zheng[3] & Jeremy R. Manning [1] ✉

Unlike temporally symmetric inferences about simple sequences, inferences about our own lives are asymmetric: we are better able to infer the past than the future, since we remember our past but not our future. Here we explore whether there are asymmetries in inferences about the unobserved pasts and futures of other people's lives. In two experiments (analyses of the replication experiment were pre-registered), our participants view segments of two character-driven television dramas and write out what they think happens just before or after each just-watched segment. Participants are better at inferring unseen past (versus future) events. This asymmetry is driven by participants' reliance on characters' conversational references in the narrative, which tend to favor the past. This tendency is also replicated in a large-scale analysis of conversational references in natural conversations. Our work reveals a temporal asymmetry in how observations of other people's behaviors can inform inferences about the past and future.

What we experience in the current moment tells us about now– but what does it tell us about the past or future? And does the current moment tell us, as human observers, more about the past or about the future? One way of examining these questions is to consider highly simplified scenarios that are artificially constructed in the laboratory (e.g.,[1]). At one extreme, for deterministic sequences with known rules, knowing the current state provides the observer with sufficient information to exactly reconstruct the entire past and future history of the stimulus. At another extreme, for purely random sequences, observing the current state provides no information about the past or future.

Sequences generated by stochastic processes fall somewhere between these two extremes. For Markov processes, where each state is solely dependent on the immediately preceding state, Shannon entropy may be used to quantify the uncertainty of the past and future states, given the present state. Cover (1994)[2] showed that, for any stationary process (i.e., processes in equilibrium), Markov or otherwise, the present state provides equal information (i.e., mutual information) about past and future states (also see[3,4]). Further, there is some evidence that humans are similarly adept at inferring the most likely previous and next items in sequences governed by stochastic Markov processes[5].

Deterministic, random, and probabilistic sequences (in equilibrium) are all symmetric: the present state of these sequences is equally informative about past versus future states. In contrast, our subjective experience in everyday life is that we know more about our own past than our future (e.g.,[6]). We have memories of our past that we carry with us into the present moment, but we do not have memories of our yet-to-be-experienced future. This temporal asymmetry imposes an "arrow of time" on our subjective experience, known as the psychological arrow of time (e.g.,[7]).

Although the psychological arrow of time implies that we should be better able to infer our past than our future, how generally does this temporal asymmetry hold? And does the asymmetry hold only for our own experiences (due to our memories), or is the asymmetry a general property of any real-life event sequence? In real-world situations (and narratives) where we are equally ignorant of the past and future, as for other people's lives where we lack memories of the relevant past, are our inferences about the past and future symmetric or asymmetric? For example, imagine that you are meeting a stranger for the first time. At the moment of your meeting, you lack both memories of their past and knowledge about what they might do in the future. After that first encounter with the stranger, would you be able to more accurately or easily form inferences about what had happened in their past (retrodiction) or what will happen in their future (prediction; Fig. 1)? Or suppose you started watching a movie partway through. Again, you

[1]Dartmouth College, Hanover, NH, USA. [2]Peking University, Beijing, China. [3]Beijing Normal University, Beijing, China.
✉e-mail: jeremy.r.manning@dartmouth.edu

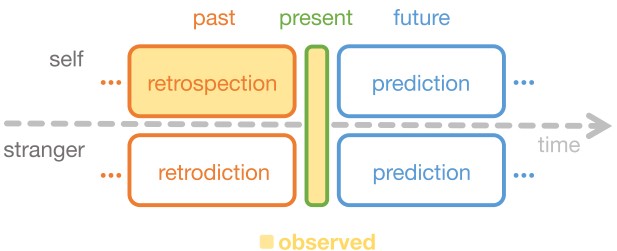

**Fig. 1 | Retrodiction, retrospection, and prediction.** In one's own life, one may draw on memory to retrospect (i.e., review or re-evaluate) the past or predict the future. This process is time-asymmetric since our own past is (typically) observed, whereas our future is not. When we make inferences about strangers' lives, however, we often have uncertainty about both their past and future, since we may have observed neither. We may retrodict the unobserved past and predict the unobserved future of strangers' lives.

would enter the moment of watching without memories of prior parts of the movie. Given your observations in the present, would your guesses about what had happened before you started watching be more (or less) accurate than your guesses about what will happen next? In general, when the past and future are both unobserved, are we better at inferring the past or the future in real-world settings? Narrative stimuli, such as stories and movies, can provide a useful testbed for exploring several of these questions.

Although narratives are unlikely to be confused with one's own experiences, narratives mirror some of the structure of real-world experiences. Character behaviors and interactions are often designed in a way that helps the audience connect with or relate to the characters. Events in narratives also unfold in ways that are intended to build rapport or engagement with the audience. This might be accomplished by having events follow a believable structure that is reminiscent of real-world experiences, or by designing the audience's experiences in ways that communicate clear "rules" or "features" that help to immerse the audience in the narrative's universe. The characters in a realistic narrative can also be written to behave in ways reminiscent of real-world people. These same aspects of narratives that authors use to drive engagement with events and characters can lead narratives to replicate some core aspects of real-world experiences that are typically lost or overlooked in traditional sequence learning paradigms. Narratives can drive the audience to build situation models[8,9] of the narrative's universe or to form a theory of mind and make predictions about the characters[10,11]. Events in narratives may unfold in a consistent or logical way, but they also exhibit complex and meaningful interactions across events reminiscent of real-world experiences (but not necessarily the simple sequences traditionally used in the statistical learning literature).

One key difference between simple artificial sequences and more naturalistic (real or narrative) sequences is that naturalistic sequences often incorporate other people. Despite the past and future being equally unknown to the observer prior to the current moment, other people and realistic characters in narratives, have their own psychological arrows of time. Specifically, they have memories of their own pasts. Other people's asymmetric knowledge about their own pasts and futures might affect their behaviors (e.g., conversations). In turn, this might provide time-asymmetric clues that favor the past (e.g., other people might talk more about their own pasts than their futures;[12]). If observers leverage these clues from other people's asymmetric knowledge, then observers should also be better at inferring the past (versus the future) of other people's lives. Alternatively, inferences about other people's lives may be more like inferences about artificial statistical sequences (e.g., perhaps solely relying on statistical regularities like event schemas, scripts, or situation models;[8,9,13–15]). If so, then the accuracy of inferences about the

past and the future of others' lives should be approximately equal. We note that the aforementioned authors make no specific claims about temporal symmetries or asymmetries. Rather, we claim that statistical regularities might imply symmetry (e.g., if you are on step $n$ of an unfolding schema, this suggests you may have just completed step $n-1$ and that you may next encounter step $n+1$).

Here, we designed a naturalistic paradigm for exposing participants to scenarios where the past and future were equally unobserved. We asked our participants to watch a series of movie segments drawn from a character-driven dramatic television show. Across the conditions and trials in the experiment, participants made free-form text responses to either retrodict what had happened in the previous segment, predict what would happen in the next segment, or recall what happened in the just-watched segment. We used manual annotations and sentence-level natural language processing models to characterize participants' responses. To foreshadow our results, we find that participants are overall better at retrodicting the past than predicting the future. This asymmetry appears to be driven by two main factors. First, characters more often refer to past events than future (e.g., planned) events, and these references are leveraged by participants to make inferences. Second, associations and dependencies between temporally adjacent events enable participants to form estimates about nearby events (e.g., a past or future event referenced in an observed conversation). We also ran a replication study with pre-registered analyses to show that these findings generalize to another television show and group of participants. Finally, we ran a large-scale analysis using natural language processing to estimate the prevalence of references to past and future events in hundreds of millions of dialogs drawn from television shows, popular movies, novels, and written and spoken natural conversations. Taken together, our work reveals a temporal asymmetry in how observations of other humans' behaviors inform us about the past versus the future.

## Results

Participants in our main experiment ($n = 36$) watched segments from two storylines, drawn from the CBS television show "Why Women Kill". Each storyline comprised 11 segments (mean duration: 2.05 min; range: 0.97–3.87 min, Supplementary Table S1). We asked participants to use free-form (typed) text responses to retrodict what had happened in the segment prior to a just-watched segment, predict what would happen in the next segment, or recall what they had just watched (Fig. 2, *Task design*). We referred to the to-be-retrodicted, to-be-predicted, or to-be-recalled segment as the target segment for each response. We systematically varied whether participants watched the segments in forward or reverse chronological order, and how many segments they had seen prior to making a response (see *Task design and procedure*). We also ran a replication study with a similar design, where participants ($n = 37$) watched segments from the Netflix television show "The Chair", comprising 13 segments (mean duration: 1.97 min; range: 0.58–4.30 min, Supplementary Table S2). The analyses of the replication experiment were pre-registered, although the replication experiment itself was not pre-registered (some of the data were collected before pre-registration).

We asked participants in our main experiment to generate four types of responses after watching each video segment: uncued responses, character-cued responses, updated responses and recalls (Fig. 2, *Data overview*). To generate uncued responses, we asked participants to either retrodict (uncued retrodiction; u-R) what happened shortly before or predict (uncued prediction; u-P) what happened shortly after the just-watched segment. To generate character-cued responses, we asked participants to retrodict (character-cued retrodiction; c-R) or predict (character-cued prediction; c-P) what came before or after the just-watched segment, but we provided additional information to the participant about which character(s) would be present in the target (to-be-retrodicted or to-be-predicted) segment. We hypothesized that character-cued responses should be more

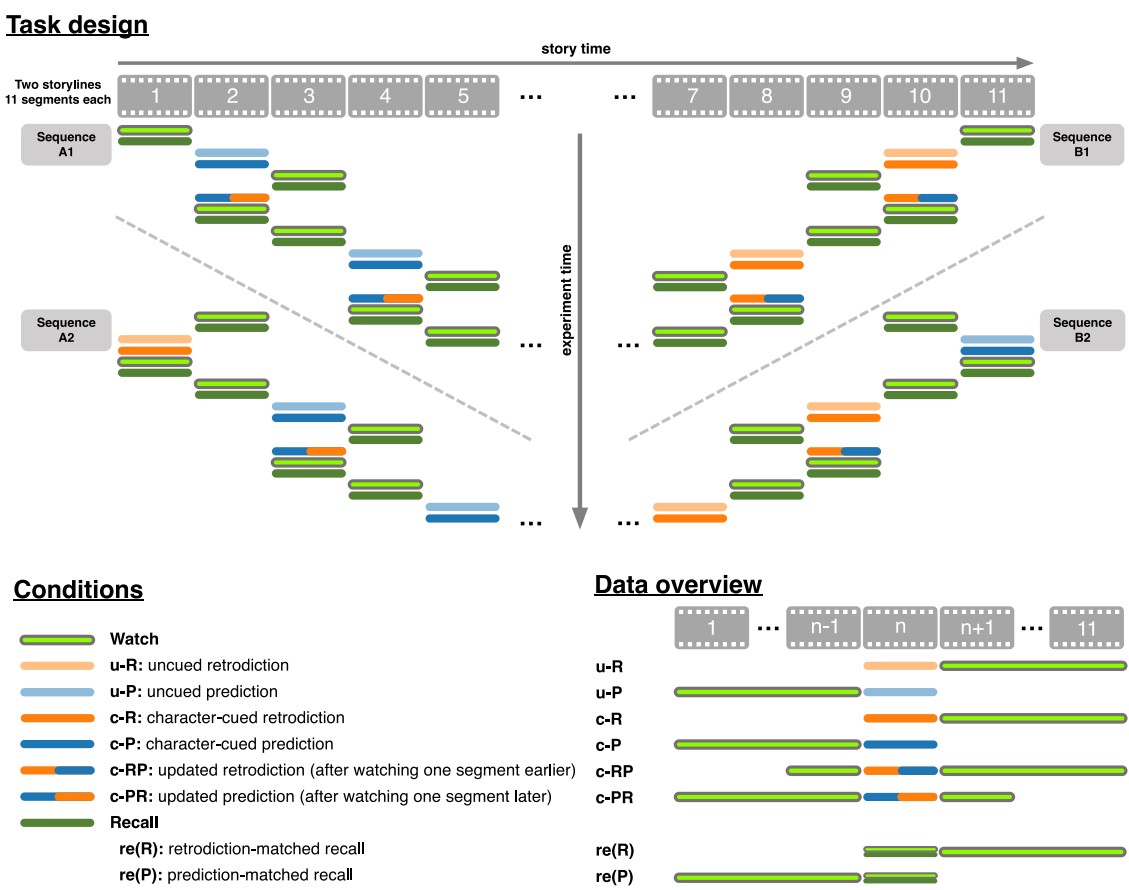

**Fig. 2 | Task overview.** Participants in our main experiment watched segments of two storylines from the television series "Why Women Kill". They made free-form text responses to either retrodict what had happened in the previous segment, predict what would happen in the next segment, or recall what happened in the just-watched segment. Across four counterbalanced sequences, we systematically varied whether participants watched the segments in forward or reverse chronological order, whether (or not) responses were cued using the main characters in the target segment, and which other segments participants had watched prior to making a response. For each segment, we collected several retrodiction, prediction, and/or recall responses across different experimental conditions. Experiment time is denoted along the vertical axis, storyline segment orders are indicated along the horizontal axis, and the colors denote experimental tasks (conditions). For an analogous depiction of our replication experiment's design, see Supplementary Fig. S1.

accurate than uncued responses, to the extent that participants incorporate the character information we provided to them into their retrodictions and predictions. To generate updated responses, we asked participants to watch an additional segment that came just prior to or just after the target segment and then to update their retrodiction (c-RP) or prediction (c-PR) about the target segment. The updated retrodiction/prediction conditions were intended to test the hypothesis that access to more information leads to better retrodiction/prediction performance. However, because this hypothesis is not directly related to the main questions about temporal symmetry we examined in this paper, we do not report on these updated responses in this manuscript. Finally, we also asked participants to recall what happened in the just-watched segment. We labeled these responses according to which other segments participants had watched prior to the just-watched target. Retrodiction-matched recall (re(R)) responses were made during the retrodiction sequences (B1 and B2; Fig. 2), whereas prediction-matched recall (re(P)) responses were made during the prediction sequences (A1 and A2; Fig. 2). Whereas retrodiction and prediction responses reflect what participants estimate they would remember after watching the (inferred) target segment, recall responses provide a benchmark for comparison by measuring what they actually remember about the target segment. Our replication experiment (Supplementary Fig. S1) used a similar design but did not have participants generate recall responses.

We used two general approaches to assess the quality of participants' responses (see *Response analyses, Text embeddings of participants' responses*, Fig. 3A). One approach entailed manually annotating events in the video and counting the number of matched events in participants' responses. We identified a total of 117 unique events reflected across the 22 video segments in our main experiment (range: 3–9 per segment), and a total of 71 events across the 13 segments in our replication experiment (range 1–16; see *Video annotation*, Supplementary Tables S1, S2). We assigned one "point" to each of these video events. We also identified a number of additional events (main experiment: 23; replication experiment: 17) in participants' responses that were either summaries of several events or that were partial matches to the manually identified video events. We assigned 0.5 points to each of these additional events. This point system enabled us to compute the numbers and proportions (hit rates) of correctly retrodicted, predicted, and recalled events contained in each response. Our second approach entailed using a natural language processing model[16] to embed annotations and responses in a 512-dimensional feature space. This approach was designed to capture conceptual overlap between responses that were not necessarily tied to specific annotated events. To quantify this conceptual overlap, we computed the similarities between the embeddings of different sets of responses. We constructed two measures, precision and convergence, to characterize different aspects of participants' responses. First, we defined

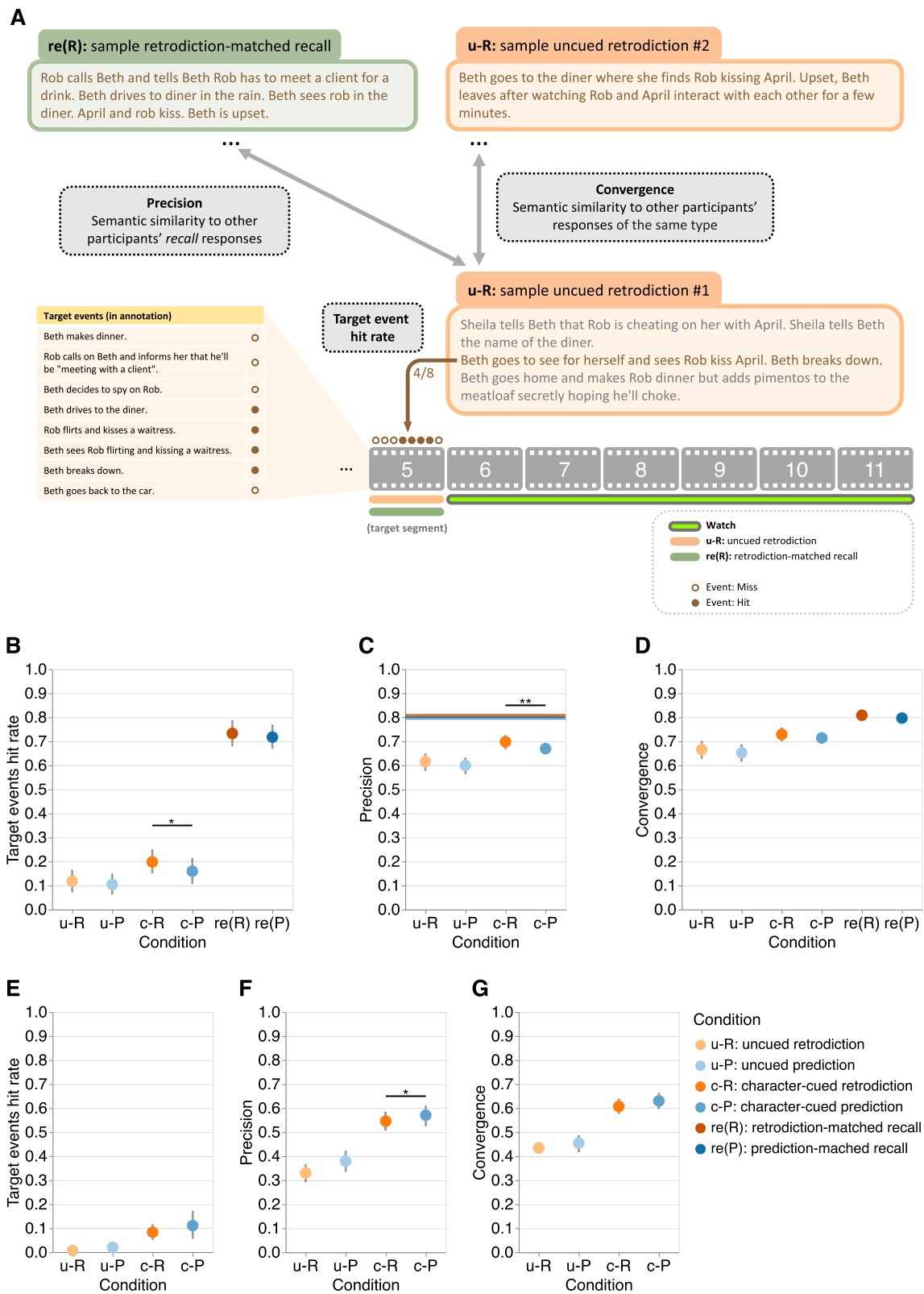

precision as the median cosine similarity between the embedding vector of a participant's retrodiction or prediction response to a target segment, and the embedding vectors for all other participants' recalls of the target segment (main experiment), or the similarity between that response's vector and the embedding vector for an online plot synopsis (obtained via Screen Spy; www.screenspy.com/the-chair-season-1-episode-1) of the target segment (replication experiment). In

this way, precision is intended to capture the degree to which participants' inferences about the target segment reflected the actual content in that segment. Next, we defined convergence as the mean cosine similarity between the embedding of a participant's retrodiction, prediction, or recall of a target segment and the embeddings of all other participants' responses (of the same type) to the same segment. In this way, convergence is intended to capture the degree of

**Fig. 3 | Retrodiction, prediction, and recall performance on target segments by experimental condition in our main and replication experiments. A Methods schematic.** For each retrodiction, prediction, and recall response, we calculated the hit rate for events in the target segment (see *Response analyses*), the response precision (see *Text embeddings of participants' responses*), and the response convergence across participants. **B Target event hit rate (main experiment).** Mean proportions of target events that were contained in participants' ($n = 36$) responses, for each response type, averaged across target segments ($n = 20$). Linear mixed models revealed no statistically significant difference between uncued retrodictions and predictions ($p = 0.729$), and higher target event hit rates in character-cued retrodictions than predictions ($p = 0.032$). **C Response precision (main experiment).** Mean precisions of participants' responses, for each response type, averaged across target segments. The horizontal lines denote the mean pairwise semantic similarities (see *Text embeddings of participants' responses*) across recall responses (re(R): orange; re(P): blue). Linear mixed models revealed no statistically significant difference between uncued retrodictions and predictions ($p = 0.287$), and higher response precision in character-cued retrodictions than predictions ($p = 0.007$. **D Response convergence (main experiment).** Mean (across-participant) convergence of participants' responses, for each response type, averaged across target segments. Linear mixed models revealed no statistically significant difference between uncued retrodictions and predictions ($p = 0.464$), and no statistically significant difference between character-cued retrodictions and predictions ($p = 0.163$). **E Target event hit rate (replication experiment).** Mean proportions of target events that were contained in participants' ($n = 37$) responses, for each response type, averaged across target segments ($n = 12$). Linear mixed models revealed no statistically significant difference between uncued retrodictions and predictions ($p = 0.054$), and no statistically significant difference between character-cued retrodictions and predictions ($p = 0.533$). **F Response precision (replication experiment).** Same format as Panel (**C**). Linear mixed models revealed no statistically significant difference between uncued retrodictions and predictions ($p = 0.083$), and lower response precision in character-cued retrodictions than predictions ($p = 0.032$). **G Response convergence (replication experiment).** Same format as Panel (**D**). Linear mixed models revealed no statistically significant difference between uncued retrodictions and predictions ($p = 0.592$), and no statistically significant difference between character-cued retrodictions and predictions ($p = 0.097$). All panels: error bars denote bootstrapped 95% confidence intervals. Asterisks indicate significance in the (generalized) linear mixed models: * denotes $p < 0.05$, and ** denotes $p < 0.01$.

similarity across participants' responses of a given type, to the given target segment. We analyzed the data using generalized linear mixed models, with participant and stimulus (e.g., target segment) identities as crossed random effects (see *Statistical analysis*).

## No evidence that retrodiction and prediction performance on target events exhibit temporal asymmetry

First, we sought to validate the main effect of response type (i.e., uncued responses, character-cued responses, and recalls), irrespective of the temporal direction (retrodiction versus prediction). Across these three types of responses, participants have access to increasing amounts of information about the target segment. Therefore, across these response types, we hypothesized that participants' responses should become both more accurate and more convergent across individuals. Consistent with this hypothesis, participants' character-cued retrodictions and predictions were associated with higher target event hit rates than uncued retrodictions and predictions in our main experiment (odds ratio (OR): 2.65, $Z = 4.24$, $p < 0.001$, 95% confidence interval (CI): 1.69 to 4.16; Fig. 3B). These character-cued responses were also more precise ($b = 0.13$, $t(18.1) = 9.43$, $p < 0.001$, CI: 0.10 to 0.16; Fig. 3C) and convergent across individuals ($b = 0.11$, $t(18.6) = 6.21$, $p < 0.001$, CI: 0.07 to 0.15; Fig. 3D). Relative to character-cued responses, participants' recalls showed higher target event hit rates (OR = 21.83, $Z = 10.61$, $p < 0.001$, CI: 12.35 to 38.59) and were more convergent across individuals ($b = 0.20$, $t(19.4) = 9.10$, $p < 0.001$, CI: 0.16 to 0.25). These results are consistent with the common-sense notion that access to more information about a target segment yields better performance (i.e., higher hit rates, precision, and convergence across individuals). These findings also held for our replication experiment (target event hit rates of character-cued vs. uncued responses: OR: 18.63, $Z = 4.26$, $p < 0.001$, CI: 4.85 to 71.58, Fig. 3E; precisions of character-cued vs. uncued responses: $b = 0.26$, $t(11.70) = 9.87$, $p < 0.001$, CI: 0.20 to 0.31, Fig. 3F; convergence of character-cued vs. uncued responses: $b = 0.25$, $t(11.98) = 8.93$, $p < 0.001$, CI: 0.19 to 0.31, Fig. 3G).

Next, we carried out a series of analyses specifically aimed at characterizing temporal direction effects− i.e., the relative quality of retrodictions versus predictions across different types of responses. We hoped that these analyses might provide insights into our central question about whether inferences about the past and future are equally accurate. Across both uncued and character-cued responses in our main experiment (Fig. 2), retrodictions had numerically higher hit rates than predictions (Fig. 3B). However, these differences were only statistically significant for character-cued responses (uncued responses: OR = 1.17, $Z = 0.35$, $p = 0.729$, CI: 0.47 to 2.92; character-cued

responses: OR = 1.93, $Z = 2.15$, $p = 0.032$, CI: 1.06 to 3.52). We observed a similar pattern of results for the precisions of participants' responses (Fig. 3C). Specifically, their responses tended to be numerically more precise for retrodictions versus predictions, but the differences were only statistically significant for character-cued responses (uncued responses: $b = 0.03$, $t(20.9) = 1.09$, $p = 0.287$, CI: − 0.03 to 0.10; character-cued responses: $b = 0.06$, $t(20.8) = 3.01$, $p = 0.007$, CI: 0.02 to 0.11). We also consistently observed numerically higher convergence across participants for retrodictions versus predictions (Fig. 3D), but neither of these differences was statistically significant (uncued responses: $b = 0.03$, $t(17.9) = 0.75$, $p = 0.464$, CI: -0.05 to 0.11; character-cued responses: $b = 0.04$, $t(17.4) = 1.46$, $p = 0.163$, CI: -0.02 to 0.09). In our replication experiment, as in our main experiment, most of these differences were not statistically significant (target event hit rates for uncued responses: OR = 0.11, $Z = − 1.92$, $p = 0.054$, CI: 0.01 to 1.04; target event hit rates for character-cued responses: OR = 1.42, $Z = 0.62$, $p = 0.533$, CI: 0.47 to 4.23, Fig. 3E; precision for uncued response: $b = − 0.06$, $t(15.86) = − 1.85$, $p = 0.083$, CI: − 0.12 to 0.01; precision for character-cued responses: $b = − 0.04$, $t(25.02) = − 2.28$, $p = 0.032$, CI: -0.08 to 0.00, Fig. 3F; convergence for uncued responses: $b = − 0.03$, $t(12.15) = − 0.55$, $p = 0.592$, CI: − 0.13 to 0.07; convergence for character-cued responses: $b = − 0.05$, $t(13.68) = − 1.78$, $p = 0.097$, CI:− 0.11 to 0.01, Fig. 3G). Taken together, our results from both experiments suggest that when we focus our analyses solely on the target segments, there is no statistical evidence that there are asymmetries in participants' inferences about the past and future.

## Retrodiction and prediction performance on events across temporal distances exhibit temporal asymmetry

The above analyses were focused solely on the target segment (i.e., retrodiction of segment $n$ after watching segments $(n + 1)…N$, or prediction of segment $n$ after watching segments $1…(n − 1)$). We wondered whether participants' responses might also contain information about more temporally distant events beyond the target segment. In order to carry out this analysis properly, we reasoned that participants might reference past or future events that were implied to have occurred offscreen but not explicitly shown onscreen. For example, a character in location A during one scene might appear in location B during the immediately following scene. Although it wasn't shown onscreen, we can infer that the character traveled between locations A and B sometime between the time intervals separating the scenes[17]. In all, we manually identified a set of 74 implicit offscreen events in our main experiment's stimuli that were implied to have occurred given what was (explicitly) depicted onscreen (Fig. 4A), plus one additional partial event and one additional summary event. We applied the same

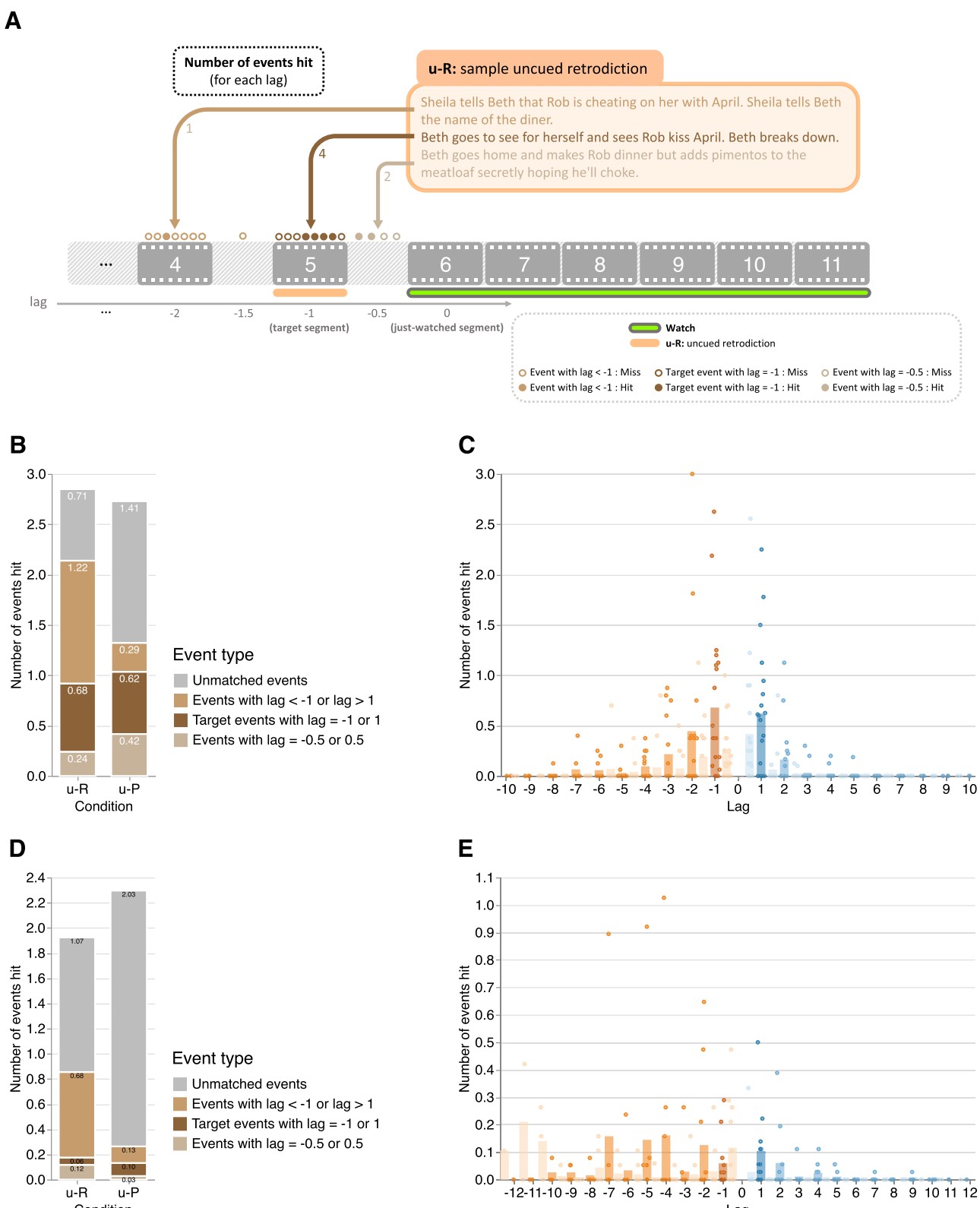

procedure to our replication experiment's stimuli and identified 66 implicit offscreen events, plus two additional partial events and one additional summary event. We defined the just-watched segment as having a lag of 0. We assigned the target segment of a participant's retrodiction or prediction (i.e., the immediately preceding or proceeding segment) a lag of −1 or +1, respectively. The segment following the next was assigned a lag of +2, and so on. We tagged offscreen events using half steps. For example, an offscreen event that occurred after the prior segment but before the just-watched segment would be assigned a lag of −0.5.

Because there is no "ground truth" number of offscreen events, we could not compute the hit rates for offscreen events. Instead, we counted up the absolute number of retrodicted or predicted events as a function of lag. In other words, given that the participant had just watched segment $i$, we asked how many events from segment $i + lag$ they retrodicted or predicted, on average, given that they were aiming

**Fig. 4 | Retrodictions and predictions of events at different temporal distances. A Illustration of the annotation approach.** For each uncued retrodiction and prediction response, we calculated the number of (retrodicted or predicted) events as a function of temporal distance from the target segment, or lag. Onscreen (explicit) events are tagged using integer-valued lags, whereas offscreen (implicit) events are tagged using half-step lags (± 0.5, ± 1.5, etc.). **B Number of events hit in participants' ($n = 36$) uncued retrodictions and predictions for each event type (main experiment).** Here we separated events we identified in participants' responses according to whether they occurred in the target segment (lags of ± 1), during the interval between the target segment and the just-watched segment (lags of ± 0.5), at longer temporal distances (|lag| > 1), or were incorrect (unmatched with any past or future events in the narrative). The counts displayed in the panel are averaged across just-watched segments. **C Number of events hit as a function of temporal distance (main experiment).** Here the (across-segment) mean numbers of events hit in participants' uncued retrodictions (orange) and predictions (blue) are displayed as a function of temporal distance to the just-watched segment (lag). Each point represents one segment (paired with a just-watched segment). **D Number of events hit in participants' ($n = 37$) uncued retrodictions and predictions for each event type (replication experiment).** Same format as Panel (**B**). **E Number of events hit as a function of temporal distance (replication experiment).** Same format as Panel (**C**). Colors denote temporal direction (orange: past; blue: future) and distance (darkest shading: target events; darker shading: onscreen non-target events; lighter shading: offscreen events).

to retrodict or predict events at lags of ± 1. We also counted the number of unmatched events in participants' responses that did not correspond to any events in the relevant segments of the narrative. We focused specifically on uncued retrodictions and predictions, which we hypothesized would provide the cleanest characterizations of participants' initial estimates of the unobserved past and future (i.e., without potential biases introduced by additional character information, as in the character-cued responses).

There were no statistically significant differences in the total (combining matched and unmatched) numbers of uncued retrodicted versus predicted events (main experiment: Ratio = 1.05, $Z = 0.75$, $p = 0.454$, CI: 0.93 to 1.18; Fig. 4B; replication experiment: Ratio = 0.93, $Z = -0.67$, $p = 0.502$, CI: 0.75 to 1.15; Fig. 4D; for results from the character-cued conditions, see Supplementary Figs. S4, S5). However, when we categorized events in participants' responses as either matched (to an event with any lag) or unmatched, we found that there were more matched events in retrodictions than in predictions (main experiment: Ratio = 1.61, $Z = 2.80$, $p = 0.005$, CI: 1.15 to 2.25; Fig. 4B; replication experiment: Ratio = 4.00, $Z = 4.33$, $p < 0.001$, CI: 2.14 to 7.50; Fig. 4D). We also observed that there were fewer unmatched events (i.e., events that did not correspond to any explicit or implicit event in the story) in retrodictions than in predictions (main experiment: Ratio = 0.34, $Z = -5.05$, $p < 0.001$, CI: 0.23 to 0.52; replication experiment: Ratio = 0.52, $Z = -4.30$, $p < 0.001$, CI: 0.39 to 0.70), suggesting that participants did not hit more events in their retrodictions solely because they made more guesses when retrodicting than predicting. We then separate matched events according to their lags. Consistent with the hit rates results reported above, the numbers of uncued retrodicted and predicted target (lag = ± 1) events were not statistically significantly different (main experiment: Ratio = 0.92, $Z = -0.15$, $p = 0.879$, CI: 0.30 to 2.84; Fig. 4B; replication experiment: Ratio = 0.44, $Z = -1.38$, $p = 0.169$, CI: 0.14 to 1.41; Fig. 4D). However, when retrodicting, participants in both experiments mentioned events with lag < −1 more often than participants predicted events with lag > 1; main experiment: Ratio = 9.10, $Z = 3.80$, $p < 0.001$, CI: 2.92 to 28.39; Fig. 4B, C; replication experiment: Ratio = 7.98, $Z = 5.50$, $p < 0.001$, CI: 3.81 to 16.74; Fig. 4D, E). We did not find any statistically significant differences in the numbers of offscreen events immediately before or after the just-watched segment in our main experiment (lag = ± 0.5; main experiment: Ratio = 0.75, $Z = -0.36$, $p = 0.715$, CI: 0.15 to 3.59), but participants in our replication experiment responded with more prior (versus future) immediately adjacent offscreen events (Ratio = 26.46, $Z = 2.45$, $p = 0.014$, CI: 1.93 to 362.29). This retrodiction advantage in the numbers of matched events held when controlling for absolute lag in our main experiment (Ratio = 34.31, $Z = 3.28$, $p = 0.001$, CI: 4.16 to 283.20), although it did not hold up in our replication experiment (Ratio > 9999, $Z = 0.00$, $p = 0.997$), as participants in the replication experiment almost never referenced offscreen events in their predictions. The retrodiction advantage also held for onscreen events alone in our main experiment (Ratio = 47.54, $Z = 3.74$, $p < 0.001$, CI: 6.27 to 360.60), but not statistically significant in our replication experiment (Ratio = 3.86, $Z = 1.86$, $p = 0.063$, CI: 0.93 to 15.98), nor for offscreen events alone (main experiment: Ratio = 24.76, $Z = 1.71$, $p = 0.087$, CI: 0.63 to 975.27; replication experiment: Ratio > 9999, $Z = 0.00$, $p = 0.997$). Again, one reason for the statistically non-significant effects in our replication experiment may be the lack of any offscreen event responses in participants' predictions. Taken together, these analyses show that participants retrodict past events with higher accuracy than they predict future events.

## The temporal bias in conversational references drives participants' asymmetric retrodiction and prediction performance

What might be driving participants to retrodict further and more accurately into the unobserved past, compared with their predictions of the unobserved future? By inspecting the video content, we noticed that characters frequently referenced both past events and (planned or predicted) future events in their spoken conversations, which might provide clues about past and future events. We wondered whether participants' responses might be influenced by characters' conversational references. Across all of the characters' conversations, and across all of the video segments from our main experiment, we manually identified a total of 82 references to past or future events (i.e., that occurred onscreen or offscreen before or after the events depicted in the current segment; Figs. 5A and Supplementary Fig. S6A, also see *Reference coding*). Characters in our main experiment's stimulus tended to reference the past (52 references) more than the future (30 references), consistent with previous work[12]. References to the past were also skewed to more temporally distant events compared with references to the future (Figs. 5B and Supplementary Fig. S6B). These asymmetries also held for characters in the replication experiment's stimulus (46 past references versus 7 future references, Supplementary Figs. S8A, S7B). These observations indicate that the characters in the stimuli display a "preference" for the past (versus future) in their conversations. Might this asymmetry be driving the asymmetries in participants' retrodictions versus predictions?

Controlling for temporal distance (lag), past and future events that story characters referenced in their conversations were associated with higher hit rates than unreferenced events in our main experiment (uncued retrodiction: OR = 12.70, $Z = 10.94$, $p < 0.001$, CI: 8.06 to 20.03; uncued prediction: OR = 8.29, $Z = 6.83$, $p < 0.001$, CI: 4.52 to 15.20; Fig. 5E). In our replication study this result held for past events (uncued retrodiction: OR = 5.57, $Z = 5.88$, $p < 0.001$, CI: 3.14 to 9.89) but not for future events (uncued prediction: OR = 1.54, $Z = 0.22$, $p = 0.827$, CI: 0.03 to 73.36; Supplementary Fig. S8D). The failure to replicate the "prediction" result appeared to follow from the fact that references to future events in characters' conversations were very rare in our replication experiment's stimulus. These findings suggest that participants' responses are at least partially influenced by the characters' conversations. To estimate the contributions of characters' references on hit rates, we computed the difference in hit rates between all events (which comprised both referenced and unreferenced events) and unreferenced events, as a function of lag. These differences exhibited a temporal asymmetry in favor of retrodiction (Fig. 5C and Supplementary Fig. S8B). This indicates that the asymmetries in participants'

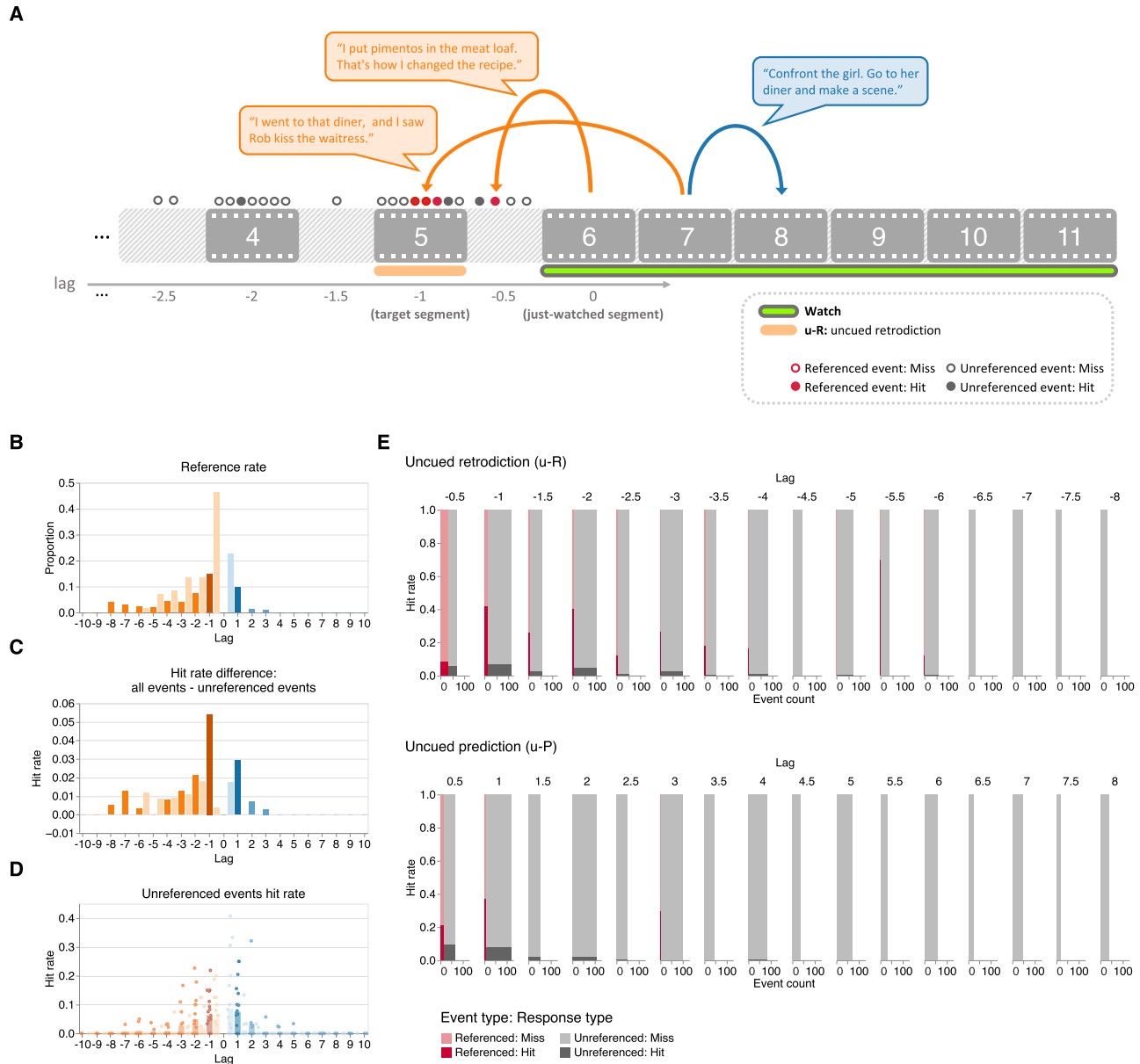

**Fig. 5 | The temporal bias in conversational references drives participants' asymmetric retrodiction and prediction performance (main experiment). A Illustration of the annotation approach.** We manually annotated references to events in past or future segments in characters' spoken conversations. We matched each such reference with its corresponding storyline event (and its corresponding segment number for onscreen events, or half-step segment number for offscreen events). We then tracked the hit rate separately for referenced versus unreferenced events in participants' uncued retrodictions and predictions. **B Reference rate as a function of lag.** Across all possible just-watched segments (lag 0), the bar heights denote the average proportions of events referenced in other past or future segments. **C Difference in hit rates between all events and unreferenced events.** To highlight the effect of characters' references to past and future events on participants' retrodictions and predictions, here we display the difference in across-segment mean hit rates between all events and unreferenced events, as a function of temporal distance (lag) to the just-watched segment. **D Hit rates for**
unreferenced events. The average response hit rates for unreferenced events are displayed as a function of temporal distance to the just-watched segment. Each point represents one segment (paired with a just-watched segment). Panels (**B**–**D**): colors are described in the Fig. 4 caption. **E Hit rates and counts of referenced and unreferenced events.** As a function of temporal distance to the just-watched segment, the sub-panels display the across-segment mean numbers (*x*-axes) and hit rates (*y*-axes) of referenced (red) and unreferenced (gray) events that participants hit (darker shading) or missed (lighter shading) in their uncued retrodictions (top sub-panel) and uncued predictions (bottom sub-panel). Intuitively, the widths of the rectangles at each lag denote the total number of events at each possible lag. The darker shading denotes the proportions of events that participants retrodicted or predicted, and the lighter shading denotes the proportions of events that participants "missed" in their responses. For an analogous presentation of results from the replication experiment, see Supplementary Fig. S8.

retrodictions versus predictions are also at least partially influenced by the characters' conversations. However, these temporal asymmetries in participants' retrodictions and predictions persisted even for events that characters never referenced in their conversations in both our main experiment (hit rates of uncued retrodicted versus predicted unreferenced events: OR = 2.00, $Z$ = 2.40, $p$ = 0.016, CI: 1.14 to 3.51;

Fig. 5D) and replication experiment (OR = 3.67, $Z$ = 2.61, $p$ = 0.009, CI: 1.38 to 9.74; Supplementary Fig. S8C). When we further separated the unreferenced events into onscreen events and offscreen events, we found that these asymmetries held only for the onscreen events in our main experiment (onscreen: OR = 2.65, $Z$ = 2.59, $p$ = 0.010, CI: 1.27 to 5.54; offscreen: OR = 1.50, $Z$ = 0.91, $p$ = 0.361, CI: 0.63 to 3.62), and only

for offscreen events in our replication experiment (onscreen: OR = 0.97, $Z = -0.06$, $p = 0.950$, CI: 0.37 to 2.57; offscreen: OR = 13.88, $Z = 3.06$, $p = 0.002$, CI: 2.58 to 74.81). Taken together, these analyses suggest that asymmetries in the number of references characters make to past and future events partially explain why participants tend to retrodict the past better than they predict the future.

## Reference-adjacent events also contribute to the temporal asymmetry

If characters' direct references cannot fully account for the temporal asymmetry in retrodicting the unobserved past versus predicting the unobserved future, what other factors might explain this phenomenon? The results above indicate that characters' references to specific unobserved events in the past or future boost participants' estimates of these events. But might characters' references have other effects on participants' responses beyond the referenced events? For example, real-world experiences and events in realistic narratives are often characterized by temporal autocorrelations (i.e., what is "happening now" will likely relate to what happens "a moment from now," and so on). Real-world experiences and realistic narratives are also often structured into "schemas" whereby experiences unfold according to a predictable pattern or formula that characterizes a particular situation, such as going to a restaurant or catching a flight at the airport[15]. If there are associations or temporal dependencies between temporally nearby events in the television shows participants watched, participants might be able to pick up on these patterns in forming their responses. This would be reflected in an inference "boost" for events that were nearby in time to events that characters referred to in their conversations, in addition to the referenced events themselves (Fig. 6A).

Because characters tended to refer to past events more often than future events, the proportions of unreferenced events that were adjacent to referenced events should show a similar temporal asymmetry in favor of the past. We tested this intuition by computing the proportions of unreferenced events in the stimulus that were temporally adjacent to past or future events referenced by the characters during a given segment. Here, we defined "temporally adjacent" as any event within an absolute lag of one relative to a referenced onscreen event, or within an absolute lag of 0.5 to a referenced offscreen event. We also defined "remaining" events as unreferenced events that were not temporally adjacent to any referenced events. In our main experiment we observed higher proportions of unreferenced past than future events that were temporally adjacent to referenced events (Fig. 6B). Further, these reference-adjacent events had higher hit rates than remaining events after controlling for absolute lag (uncued retrodiction: OR = 7.15, $Z = 2.40$, $p = 0.016$, CI: 1.44 to 35.58; uncued prediction: OR = 3.11, $Z = 2.30$, $p = 0.022$, CI: 1.18 to 8.21; Fig. 6E). To estimate the contributions of reference adjacency on hit rates, we computed the differences in hit rates between unreferenced events (which comprised both reference-adjacent and remaining events) and remaining events, as a function of lag. These differences exhibited a temporal asymmetry in favor of retrodiction (Fig. 6C). This suggests that reference-adjacent events also contribute to participants' retrodiction advantage. This reference-adjacency effect did not hold in our replication experiment (uncued retrodiction: OR = 6.46, $Z = 1.58$, $p = 0.113$, CI: 0.64 to 65.04; uncued prediction: OR = 0.002, $Z = 0.007$, $p = 0.995$, CI: < 0.001 to > 9999; Supplementary Fig. S9D, B). Upon further examination of the stimulus we used in our replication experiment, along with participants' responses, we noticed that the television episode appears to comprise several interleaved storylines (Supplementary Fig. S10A). This meant that what we had originally labeled as "reference-adjacent" events (based solely on the temporal order in the episode) did not necessarily correspond to the chronological order in the story. For example, if (across successive segments) the narrative focuses on character $A$ at time $t$ in segment $n$, and

on character $B$ at time $t$ in segment $n + 1$, then we reasoned that watching segment $n$ might not provide much insight into what would happen in segment $n + 1$. However, watching segment $n$ could provide clues about what would happen to character $A$ at time $t + 1$, which might have been shown later on in the episode. We then corrected the reference-adjacency labels in the replication experiment stimulus to correspond to individual storylines, rather than solely with respect to the episode segment orders. This correction was not pre-registered. With this correction, we recovered the reference adjacent effect for uncued retrodiction (OR = 7.55, $Z = 2.93$, $p = 0.003$, CI: 1.95 to 29.20; Supplementary Fig. S10E, C). We did not find a significant reference adjacent effect in uncued prediction (OR = 1176.66, $Z = 0.04$, $p = 0.972$, CI: < 0.001 to > 9999), again likely due to the limited number of future references in the narrative. The remaining events did not exhibit a statistically significant temporal asymmetry (main experiment: OR = 0.75, $Z = 0.33$, $p = 0.739$, CI: 0.14 to 4.08; Fig. 6D; replication experiment: OR = 889.48, $Z = 0.03$, $p = 0.973$, CI: < 0.001 to > 9999; Supplementary Fig. S10D). These results suggest that temporal adjacency can also partially account for participants' retrodiction advantages.

## Effects of references on retrodiction and prediction performance are directed

The preceding analyses show that when characters reference past or future events, those referenced events and other events that are temporally adjacent to the referenced events are more likely to be retrodicted and predicted. In other words, referring to a past or future event in conversation leads to a "boost" in that event's hit rate. We wondered whether this boost was bi-directional. In particular: when a character refers (during a referring event) to another event (i.e., the referenced event), does this boost only the referenced event's hit rate, or does the referring event also receive a boost? We labeled each event as a "referring event," a "referenced event," or an "other event" (i.e., not referring or referenced; Fig. 7A, B). We limited our analysis to references to onscreen (explicit) events. Consistent with our analysis of the proportions of referenced events (Fig. 5B and Supplementary Fig. S8A), the proportions of referring events exhibited a forward temporal asymmetry (Fig. 7C and Supplementary Fig. S11A). Controlling for absolute lag, we found that referring events were associated with lower hit rates than referenced events in our main experiment (uncued retrodiction: OR = 0.02, $Z = -4.12$, $p < 0.001$, CI: 0.00 to 0.11; uncued prediction: OR = 0.04, $Z = -4.80$, $p < 0.001$, CI: 0.01 to 0.15; Fig. 7D) and had no statistically significant differences in hit rates compared with other events (uncued retrodiction: OR = 0.37, $Z = -1.07$, $p = 0.284$, CI: 0.06 to 2.28; uncued prediction: OR = 2.33, $Z = 1.53$, $p = 0.127$, CI: 0.79 to 6.87). In our replication experiment, because there were very few referenced events during prediction, which also resulted in limited referring events during retrodiction, we had insufficient data to reliably compare between referenced, referring, and other events (all $p$s > 0.998; Supplementary Fig. S11). Taken together, this indicates that only referenced events received a hit rate boost (relative to other events), suggesting that the retrodictive and predictive benefits of references are directed (i.e., asymmetric).

## Large-scale analysis of conversational data replicates the temporal bias in conversational references

The above analyses show that participants leveraged characters' references to make inferences about the past and the future, and the retrodiction advantage could be attributed to the fact that characters in the television shows we used as stimuli in our main experiment and replication experiment refer more often to the past than to the future. But how universal is this pattern? For example, were the television shows we happened to select for our experiments representative of television shows more generally? Or perhaps narratives created for entertainment purposes tend to have biases towards the past in order

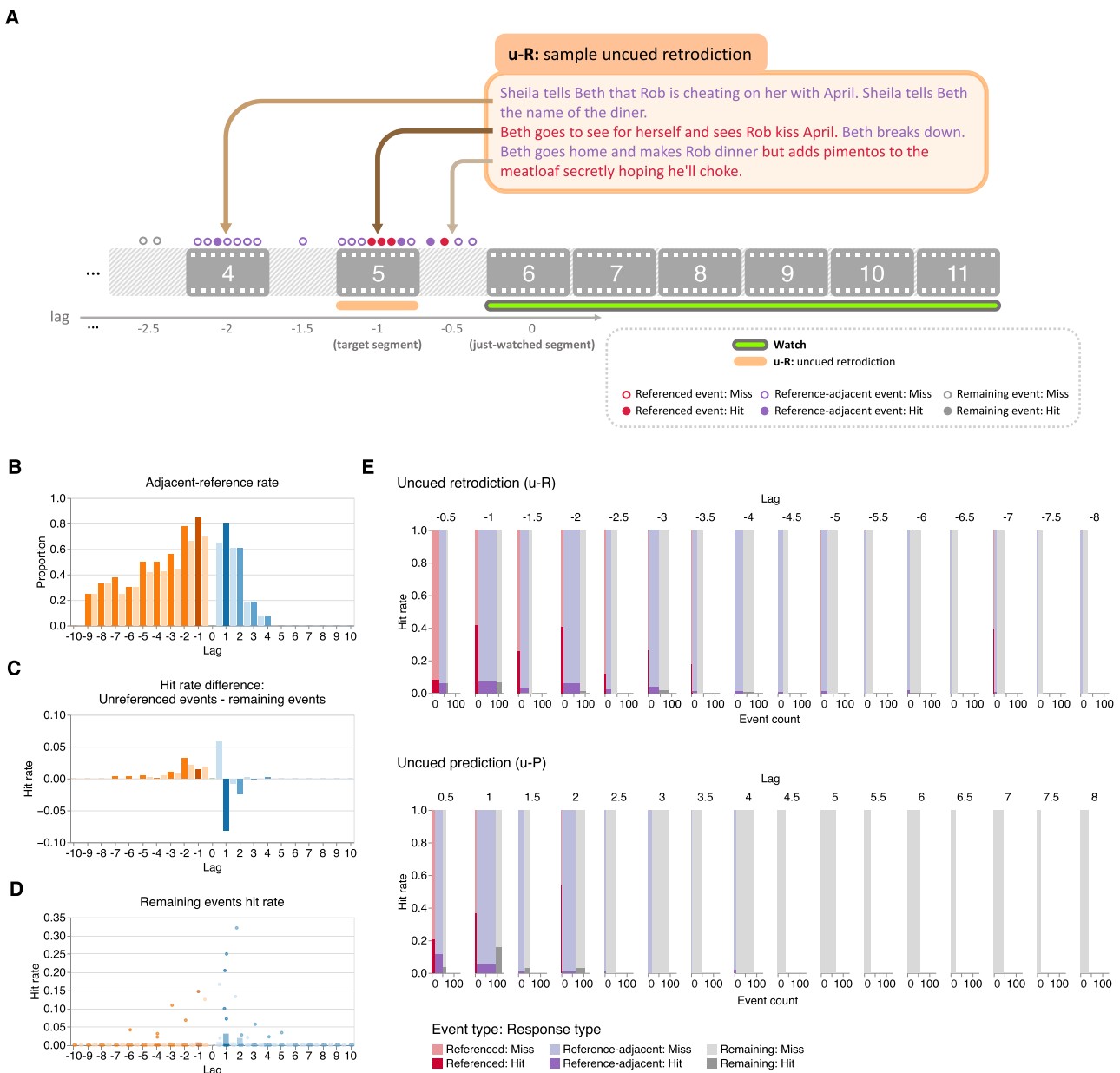

**Fig. 6 | Reference-adjacent events also contribute to the temporal asymmetry (main experiment). A Illustration of the annotation approach.** We extended the annotation procedure depicted in Fig. 5A to also label unreferenced events that were either temporally adjacent to (i.e., immediately preceding or proceeding) a referenced event (reference-adjacent events) or not (remaining events). **B Adjacent reference rate for unreferenced events as a function of lag.** Across all possible just-watched segments (lag 0), the bar heights denote the average proportion of unreferenced events in other past or future segments that were temporally adjacent to any referenced event. **C Difference in hit rates between unreferenced events and remaining events.** To highlight the effect of reference adjacency on retrodiction and prediction of unreferenced events, here we display the difference in across-segment mean hit rates between unreferenced events and remaining events, as a function of temporal distance (lag) to the just-watched segment. **D Hit rates for remaining events.** Participants ($n = 36$) across-segment mean response hit rates for unreferenced events that were not temporally adjacent to any referenced events are displayed as a function of temporal distance to the just-watched segment. Each point represents one segment (paired with a just-watched segment). Panels (**B**–**D**): colors are described in the Fig. 4 caption. **E Hit rates and counts of referenced, reference-adjacent, and remaining events.** As a function of temporal distance to the just-watched segment, the sub-panels display the numbers (x-axes) and proportions (y-axes) of referenced (red), reference-adjacent (purple), and remaining (gray) events that participants hit (darker shading) or missed (lighter shading) in their uncued retrodictions (top sub-panel) and uncued predictions (bottom sub-panel). For an analogous depiction of results from our replication experiment, see Supplementary Figs. S9, S10.

---

to keep the stories engaging and unpredictable. To better understand temporal biases in conversations, we carried out a large-scale analysis using extracted conversation data from 12 datasets, comprising over 17 million documents. The data comprised transcripts from television shows and popular films, novels, and spoken and written utterances from natural conversations. These datasets were chosen partly for convenience, and partly to sample from a variety of different types of

conversations. Nine of the datasets we examined were from ConvoKit (https://convokit.cornell.edu/documentation/datasets.html;[18]). Using the category labels proposed by ref. [19], the conversations we analyzed included "constrained," "scripted," and "spontaneous" human-human conversations. A summary of the data we analyzed, including brief descriptions of each dataset, may be found in Supplementary Table S3. We used natural language processing to identify

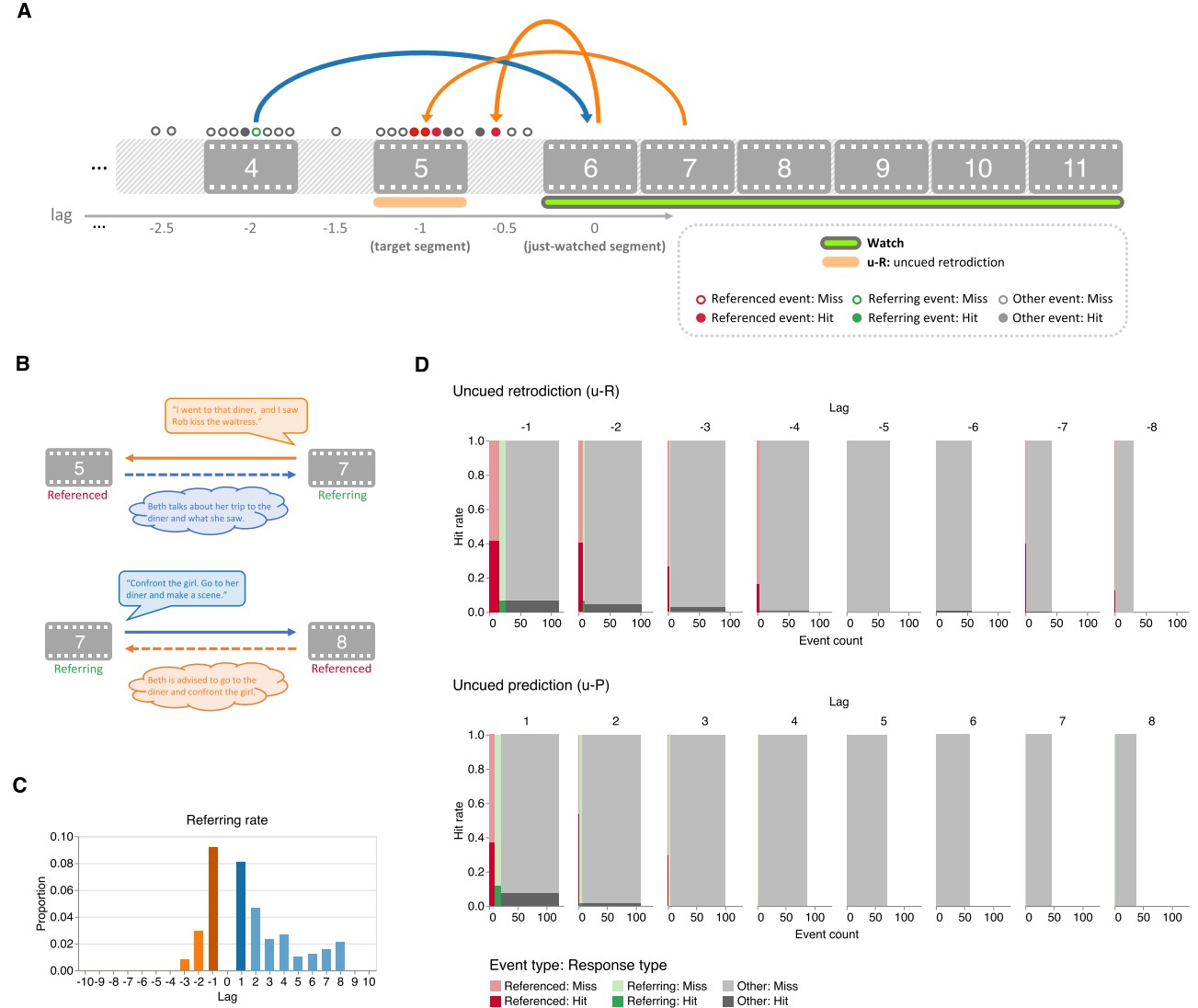

**Fig. 7 | Effects of references on retrodiction and prediction performance are directed (main experiment). A Illustration of the annotation approach.** We extended the annotation procedure depicted in Fig. 5A to also label which events in our main experiment's stimuli contained references to events in other segments. **B Referenced versus referring events.** During event *i*, when a character makes a reference to another event (*j*), we define *i* as the "referring" event and *j* as the "referenced" event. **C Refers to the rate as a function of lag.** Across all possible just-watched segments (lag 0), the bar heights denote the across-segment mean proportions of events containing references to events in other past or future

segments in our main experiment's stimuli. The bar colors are described in the Fig. 4 caption. **D Hit rates and counts of referenced, referring, and other events.** As a function of temporal distance to the just-watched segment, the sub-panels display the numbers (*x*-axes) and hit rates (*y*-axes) of referenced (red), referring (green), and other (gray) events that participants hit (darker shading) or missed (lighter shading) in their uncued retrodictions (top sub-panel) and uncued predictions (bottom sub-panel). For a display of analogous results from our replication experiment, see Supplementary Fig. S11.

references to past or future events in each conversation (see *Large-scale analysis of conversational data*).

In all, across all of the datasets we examined in our analysis, we identified a total of 24,040,006 references to past or future events. A total of 13,471,984 (56.04%) of these were references to past events, and the remaining 10,568,022 (43.96%) were references to future events. For each dataset, we tested whether the proportion of references to past events was higher than that of references to future events. In 11 of the 12 datasets, there were higher proportions of references to past events than future events (all *ps* < 0.001). In one dataset, "Persuasion for Good," which comprised natural conversations between pairs of Amazon Mechanical Turk workers wherein one participant tried to convince the other participant to donate to a charity in the future, references to the future were significantly more common than references to the past ($\chi^2(1) = 422.31$, *p* < 0.001). This latter example provided a nice sanity check for verifying that our

general approach was not itself biased in favor of the past, e.g., even in conversations that were actually biased towards the future.

We used a meta-analysis-like approach to test if there were reliably more references to past events than to future events across all of the datasets (Fig. 8). For each dataset, we used as our effect size the Past/Future ratio, defined as the proportion of references to past events (in all sentences) divided by the proportion of references to future events (commonly known as the "risk ratio"). In this way, effect sizes greater than 1 reflect a bias toward the past, whereas effect sizes of less than 1 reflect a bias toward the future. We fit a random-effects model to the log-transformed ratios. For the 12 datasets we included in this analysis, the observed ratios ranged from 0.76 to 2.09, with the majority of estimates being greater than one (92%). The estimated average ratio based on the random-effects model was $\mu = 1.44$ (95% CI: 1.23 to 1.69). Therefore, the average outcome differed significantly from zero ($z = 4.57$, *p* < 0.001). According to Cook's distances, none of the studies

| Dataset | Type | Total | Past | Future | | Ratio [95% CI] |
|---------|------|-------|------|--------|---|---------------|
| IMSDb | Scripted | 3080674 | 657475 | 316525 | | 2.08 [2.07, 2.09] |
| Movies | Scripted | 516163 | 127744 | 85937 | | 1.49 [1.48, 1.50] |
| Switchboard | Spontaneous | 245461 | 41488 | 22079 | | 1.88 [1.85, 1.91] |
| SCOTUS | Constrained | 3880259 | 1963578 | 1207377 | | 1.63 [1.62, 1.63] |
| Tennis | Constrained | 599172 | 281669 | 134638 | | 2.09 [2.08, 2.10] |
| PfG | Constrained | 37184 | 7408 | 9771 | | 0.76 [0.74, 0.78] |
| IQ2 | Constrained | 122925 | 46630 | 34811 | | 1.34 [1.32, 1.35] |
| GAP | Constrained | 8009 | 1800 | 1338 | | 1.35 [1.26, 1.43] |
| Chair | Scripted | 2900 | 660 | 460 | | 1.43 [1.29, 1.60] |
| Friends | Scripted | 107082 | 22067 | 16356 | | 1.35 [1.32, 1.37] |
| Gutenberg | Scripted | 29119393 | 10234952 | 8672030 | | 1.18 [1.18, 1.18] |
| Reddit | Constrained | 217924 | 86513 | 66700 | | 1.30 [1.29, 1.31] |
| RE Model | | | | | | 1.44 [1.23, 1.69] |

Ratio (log scale): 0.67  1  1.5  2.25

**Fig. 8 | Large-scale analysis of conversational data replicates the temporal bias in conversational references.** We used natural language processing to automatically identify references to past or future events across a variety of datasets. The "type" label was applied following[19]. For each dataset, we calculated the ratio of references to past events and references to future events. We performed a large-scale analysis of the ratios with a meta-analysis-like approach using a random effects model. Data are presented as ratios ± 95% CI, with a summary estimate based on the random effects model. Dataset descriptions may be found in Supplementary Table S3.

could be considered to be overly influential. Taken together, the results from our analysis indicate that people tend to refer to the past more than they refer to the future, across a wide variety of situations (including in both fictional and real conversations). Although (as in the Persuasion for Good dataset) there may be specific exceptions to this bias, it seems that a bias in favor of the past is a common element of many (and perhaps even most) human-human conversations.

## Discussion

We asked participants in our main experiment to watch sequences of movie segments from a character-driven television drama and then either retrodict what had happened prior to a just-watched segment, predict what would happen next, or recall what they had just watched. We found that participants tended to more accurately and more readily retrodict unobserved past events than predict unobserved future events. We traced this temporal asymmetry to (a) characters' tendencies to refer to past events more than future events in their ongoing conversations, and (b) associations between temporally proximal events (Fig. 9). Essentially, associations between temporally proximal events serve to enhance asymmetries in inferences driven by conversational references (light orange and blue bars in Fig. 9). Our findings show that other peoples' psychological arrows of time can affect external observers' inferences about the unobserved past and future. In a replication study with pre-registered analyses, we

replicated our main findings of (a) an asymmetry in participants' inferences of past and future events and (b) the factors contributing to this asymmetry. We also carried out a large-scale analysis of tens of millions of utterances from television shows, movies, novels, and natural spoken and written conversations. We found that the tendency to refer more often to the past than the future appears to be a widespread characteristic of human conversation.

There exists a fundamental knowledge asymmetry such that we know more about our own past than the future since we remember our past but not our future. A number of prior studies have examined other temporal biases, such as how much people focus on the past, present, and future in their spontaneous thoughts[20–22], everyday conversations[12], and social media messages[23]. Several of these studies found that, on average, people's spontaneous internal thoughts tend to be more future-oriented than past-oriented[20,21]. In contrast, people's external communications tend to focus more on the past[12,23].

When people communicate through language or other observable behaviors, they can transmit their knowledge and memories to others[24–27]. A consequence of this sharing across people is that biases or limitations in one person's knowledge and memories may also be transmitted to external observers. Although people can communicate their intentions and future plans (i.e., information about their future), because people know more about their pasts than their futures, the knowledge transmitted to observers is inherently biased in favor of the

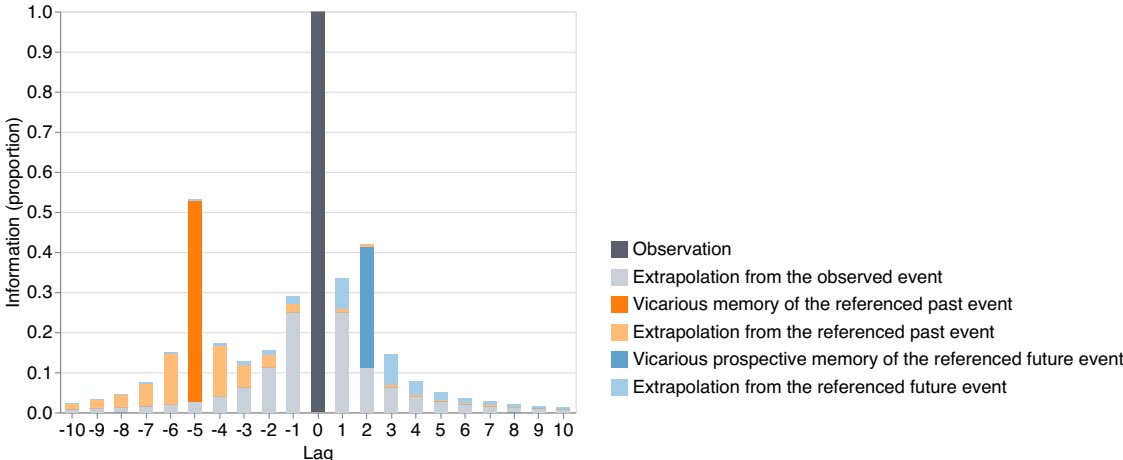

**Fig. 9 | How much information about the past and future can be inferred by observing the present?** By definition, let us say that the present moment (lag 0) contains all information about itself (dark gray). Given learned statistical regularities, one might extrapolate from the present moment into the past or future (light gray). As illustrated in this schematic, the information contained in the present about other moments in time falls off with absolute lag. This falloff is approximately time-symmetric. References in the present to past events (dark orange) or future events (dark blue) provide additional information about those referenced moments in time, beyond what could be inferred solely from statistical regularities. This additional information about those referenced moments can also be extrapolated to other moments that are temporally nearby to them (light orange and blue). The data in this schematic are hypothetical.

past (Fig. 9;[12]). Since observers leverage communicated knowledge to reconstruct the unobserved past and future, this explains why observers' inferences about observed people's lives also favor the past.

People's knowledge asymmetries are not always directly observable. For example, in a conversation where someone talks exclusively about their future plans, a passive observer might gain more insight into the speaker's unobserved future than their unobserved past. However, because the speaker is also guided by their own psychological arrow of time, the "upper limit" of knowledge about their past is still higher than that of their future. Therefore, after accounting for knowledge that could be revealed through active participation in the conversation, the seemingly future-biased conversation masks an underlying knowledge asymmetry in favor of the past. This hypothesized "unmasking" effect of interaction implies that the influence of other people's psychological arrows of time should be more robust when the receiver is an active participant in the conversation. Other social dimensions, such as trust, motivation or level of engagement, personal goals, and beliefs, might serve to modulate the effective "gain" of the communication channel– i.e., how much the speaker's knowledge influences the observer's knowledge.

In typical statistical sequences used in laboratory studies, there is no temporal asymmetry, either theoretically[2–4] or empirically[5]. What makes narratives and real-world event sequences time-asymmetric? Of course, there are many superficial differences between simple laboratory-manufactured sequences and real-world experiences. As one example, real-world experiences often involve other people who have their own memories and goals. Some recent work (e.g.,[28,29]) suggests that people might gain insights into other people using "mental simulations" of how they might respond in particular situations (e.g., in the future), or of which sorts of prior experiences might have led someone to behave a particular way in the present. But at a higher level, are our subjective experiences essentially more complicated versions of laboratory-manufactured sequences? Or are there fundamental differences? One possibility is that real-life event sequences are not stationary (i.e., not in equilibrium;[2]). For example, real-life events might start from a special low-entropy initial condition[2,30,31] and proceed through a series of transitions from low-entropy to high-entropy states, thus exhibiting an arrow time. When we retrodict, it is possible that we only consider possible past events that are compatible with the low-entropy initial state[32,33]. For example, when we see a broken egg we might infer that the egg had been intact at some point in the past. But it

would be difficult to guess what states or forms the broken egg might take in the future[32,33]. In other words, the procession from low entropy to high entropy might result in better retrodiction performance compared with that of (implicitly less-restricted) prediction tasks. The low-entropy initial state might also explain why we remember the past, but not the future. Some recent work suggests that the psychological arrow of time might be explained by a related concept in the statistical physics literature, termed the "thermodynamic" arrow of time[34,35]. However, the relation between the thermodynamic and psychological arrows of time is still under debate[36,37].

Beyond forming inferences about unobserved past and future events, our work also relates to prior studies of how people perceive time[38–42], and how we "move" through time in our memories of our past experiences[43–50] or in our imagined (past or future) experiences[51–54]. For example, a well-studied phenomenon in the episodic memory literature concerns how remembering a given event cues our memories of other events that we experienced nearby in time (i.e., the contiguity effect;[48]). Across a large number of studies, there appears to be a nearly universal tendency for people to move forward in time in their memories, whereby recalling an "event" (e.g., a word on a previously studied list) is about twice as likely to be followed by recalling the event that immediately followed as compared with the event immediately preceding the just-recalled event[55]. Superficially, our current study appears to report the opposite pattern, whereby participants display a backward temporal bias. However, the two sets of findings may be reconciled when one considers the frame of reference (and current mental context; e.g.,[56]) of the participant at the moment they make their response. In our study, participants observe an event in the present, and they make guesses about what happened in the unobserved past or future, relative to the just-observed event. (Our findings imply that participants are more facile at moving backward in time than forward in time, relative to "now.") In contrast, the classic contiguity effect in episodic memory studies refers to how people move through time relative to a just-remembered event. The forward asymmetry in the contiguity effect follows from the notion that the moment of remembering has greater contextual overlap with events after the remembered event from the past, including the moment of remembering, than events that happened before it (for review also see[57,58]). In other words, our current frame of reference appears to exhibit a sort of "pull" on our thoughts, such that thoughts about recent experiences still lingering in our minds drag us towards the

recent past, but after thinking about the more distant past, we are dragged (relatively) forward in time back to "now."

We acknowledge several limitations in the current study. First, our experimental sample sizes are limited in the number of participants and stimuli. We addressed this limitation by running a replication of our main experiment with pre-registered analyses to verify that the main behavioral findings were held in a new cohort of participants and in a new set of stimuli. We also used a meta-analysis approach to count the numbers of past versus future references in millions of natural conversations, which (in our experiments) appeared to be a major factor in participants' inferences. We found that, as in the experimental stimuli, references to the past consistently outnumbered references to the future in the conversations we examined. However, although this large-scale analysis shows that the conversational asymmetries we observed in the experimental stimuli are commonplace across many conversations, we acknowledge that we did not directly ask participants to form inferences about the past or future from observations of those conversations. Future studies could select a wider range of stimuli to see whether the retrodiction advantage holds in different scenarios. Second, while we selected narrative stimuli that seemed "realistic" in an attempt to elicit behaviors that might generalize to "real life," there are some important differences between narratives and everyday experiences in real life that are worth noting. Compared with everyday experiences, narratives might have different statistical properties. For example, narratives often incorporate deliberate causal associations across events[59] that may not be present or obvious in real experiences. In real life, we also often have access to traces of the past beyond conversational references. For example, we can rely on artifacts like photos, letters, notes, digital records, etc., to infer what likely happened in the past. Future studies could incorporate a variety of ways to study how we make inferences about the past. Another difference between narratives and real-world experiences is that we typically consume narratives through passive listening and viewing, whereas real-world experiences are interactive. An important question for future work will be to clarify the role of active participation in how we understand the present and how we form inferences about the past and future.

In our study, we explicitly designed participants' experiences such that both the past and future were unobserved. How representative is this scenario of everyday life? For example, we might try to speculate about the unobserved future when making plans or goals, but when might we encounter situations where the past is unobserved but still useful for us to speculate about? Real-life events have long-range dependencies. In general, because the future depends on what happened in the past, discovering or estimating information about the unobserved past can help us form predictions about the future. We illustrate this point in Fig. 9 by showing that the additional information contributed by a referenced past event can also extend into the future (light orange bars at lags > 0). This might explain why humans devote substantial effort and resources to attempting to figure out what happened in the unobserved past: history, anthropology, geology, detective and forensic science, and other related fields are each primarily focused on understanding, retrodicting, or reconstructing unobserved past events.

## Methods
All protocols (main experiment, replication experiment, and large-scale analysis) were approved by the Committee for the Protection of Human Subjects at Dartmouth College. Participants gave written consent to enroll in the two experiments.

### Participants
**Main experiment.** A total of 36 participants (25 female, mean age 21.47 years, range 19–50 years) were recruited from the Dartmouth College community for our main experiment. No statistical method was used to predetermine the sample size. We had no a priori hypotheses about gender differences in this study, thus we did not conduct any gender-based analysis. All participants had self-reported normal or corrected-to-normal vision, hearing, and memory, and had not watched any episodes of "Why Women Kill" before the experiment. Participants received course credit or monetary compensation for their time. Two participants completed only the first half of the study and one participant's data from the second half of their testing session was lost due to a technical error. All available data were used in the analyses.

**Replication experiment.** A total of 37 participants (21 female, mean age 22.24 years, range 19–30 years) were recruited from the Dartmouth College community for our replication experiment. No statistical method was used to predetermine the sample size. All participants had self-reported normal or corrected-to-normal vision, hearing, and memory, and had not watched any episodes of "The Chair" before the experiment. Participants received monetary compensation for their time. For two participants, one segment was not played due to a technical error, resulting in four unregistered trials. All available data were used in the analyses. Analyses of the replication experiment were pre-registered on May 24th, 2023 (https://aspredicted.org/8e4ad.pdf). Some of the data were collected before pre-registration.

### Stimuli
**Main experiment.** The stimuli used in our main experiment were segments of the CBS television series "Why Women Kill" Season 1. The TV series contained three distinct storylines depicting three women's marital relationships. The three storylines, which took place in the 1960s, 1980s, and 2019, were shown in an interleaved fashion in the original episodes. The first 11 segments from the 1960s and 1980s storylines, across the first and second episodes, were used in our study. Segments were divided based on major scene cuts, which primarily corresponded to storyline shifts in the original episodes. The mean length of the segments was 2.05 min (range 0.97–3.87 min). We chose this TV series based on its strictly linear storytelling (within each storyline) and its realistic settings where most events depict everyday life. The plots were focused on the main characters (Beth in storyline 1 and Simone in storyline 2), who were present in all the segments in the corresponding storylines.

**Replication experiment.** The stimuli used in our replication experiment were segments of the first episode of the Netflix television show "The Chair", Season 1. The TV series depicts the life of a professor who is the English department chair at a major university. The first episode was used in our study and was divided into 13 segments. Segments were divided based on major scene cuts and were minimally edited. The mean length of the segments was 1.97 min (range 0.58–4.30 min). We chose this TV series based on its strictly linear storytelling and its realistic settings where most events depicted everyday life on a college campus.

### Task design and procedure
**Main experiment.** Our experimental paradigm was divided into two testing sessions. In each session, participants performed a sequence of tasks on segments from one storyline (Fig. 2). For each storyline, there were four different task sequences: two forward chronological order sequences and two backward chronological order sequences. Participants completed one task sequence in forward chronological order for one storyline, and one in backward chronological order for the other storyline. The order of the two sessions (forward chronological order sequence first or backward chronological order sequence first), and the pairing of task sequences with storylines, were counterbalanced across participants. The experiments were not randomized.

The Investigators were not blinded to allocation during experiments and outcome assessment.

Tasks in each sequence alternated between watching, recall, and retrodiction or prediction, with the specific order of tasks differing across the four sequences. For example, in sequence A1, participants first watched segment 1, followed by an immediate recall of segment 1. Then they predicted what would happen in segment 2 (first uncued and then character-cued). Participants then watched segment 3 and recalled segment 3. After that, participants guessed what happened in segment 2 again, which we termed "updated prediction". Then they watched segment 2, recalled segment 2, and so on, as depicted in Fig. 2. This procedure was repeated to cover all possible segments. We also note several edge cases at the start and end of the narrative sequences. Since no segments precede the first segment, participants could never make "prediction" responses with the first segment as their target. For analogous reasons, participants never made "retrodiction" responses with the last segment as their target. Another edge case occurred in task sequences B2 and A2 (Fig. 2). In the A1 and A2 sequences, participants experience the narrative in the original (forward) order, predicting one segment ahead along the way. In the B1 and B2 sequences, participants experience the narrative in the reverse order, retrodicting one segment ahead along the way. However, because A2 and B2 are offset from A1 and B1 by one segment, the initial A2 responses are retrodictions, and the initial B2 responses are predictions (i.e., they conflict with the temporal directions of the remaining responses in those conditions). We, therefore, excluded from our analysis those initial retrodiction responses from the A2 condition and the initial prediction responses from the B2 condition.

Before watching each segment, participants were given the following task instructions. After watching the video, participants were instructed to type their responses (retrodiction, prediction, or recall) in 1–4 sentences. Participants were also asked to specify the characters' names in their responses, i.e., avoiding the use of characters' pronouns. For the recall task, the names of the characters in the recall segment were displayed, and participants were asked to summarize the major plot points in the present tense. For the retrodiction and prediction tasks, participants were instructed to retrodict or predict the major plot points of the segment (also in the present tense), as though they had watched the segment and were writing a plot synopsis. They were also instructed to avoid speculation words (e.g., "I think Beth will…"). For the uncued retrodiction and prediction tasks, participants made retrodictions or predictions without any cues provided, so they had to guess which of the characters would be present in the segment. For character-cued retrodictions and predictions, the characters in the target segment were revealed on the screen, alongside participants' previous responses. Participants were instructed to include or incorporate those characters into their character-cued responses if their previous responses did not contain all the characters provided. They were also told that the characters were not necessarily listed in their order of appearance in the segment and that only the main characters would be given. Also, the characters given did not necessarily interact with each other in that segment, and they could appear in successive events in that segment. If participants' previous responses included all the characters given, then they could directly proceed to the next task without updating their responses. For each retrodiction and prediction, participants were asked to generate at least one, and not more than three, responses that constituted "the sorts of things [the participant would] expect to have remembered if [they] had watched the [target] segment." They were asked to generate multiple responses only if those additional responses were (in their judgment) of equal likelihood to occur. On average, participants in our main experiment generated 1.08 responses per prompt; therefore, we chose to consider only participants' first ("most probable" or "most important") responses to each prompt. Each response (including recall) was followed by a confidence rating on a 1–5 point scale.

However, these confidence data were not analyzed in the present study.

Before their first testing session, participants were given a practice session, where they watched the first segment of storyline 3, followed by a recall trial, an uncued prediction trial, and a character-cued prediction trial. Participants' responses were checked by the experimenter to ensure compliance with the instructions. To provide participants with sufficient background information about the storyline (especially for the backward chronological sequences), at the beginning of each session, participants were shown the time, location, and the main characters (with pictures) of the storyline. The first session was approximately 1.5 hours long, and the second session was approximately 1 hour long. We allowed participants, at their own discretion and convenience, to sign up for two consecutive testing time slots (i.e., with their testing sessions occurring in immediate succession) or for testing sessions on two different days. The mean intersession interval was 0.73 days (range: 0–4 days). The experiment was conducted in a sound- and light-attenuated testing room. Videos were displayed using a 27-inch iMac desktop computer (resolution: 5120 × 2880), and sound was presented using the iMac's built-in speakers. The experiment was implemented using jsPsych[60] and JATOS[61].

**Replication experiment.** The design and procedure of the replication experiment were similar to the main experiment, other than the following differences. In the replication experiment, we used only one storyline, and therefore, participants performed only one task sequence (either chronological or backward chronological), in one session (Supplementary Fig. S1). Tasks alternated between watching, and retrodiction or prediction. Some segments contained multiple scenes with different characters. For these segments, characters for each scene were shown in the cued conditions, and participants were asked to guess what would happen in each scene between these characters. For each retrodiction and prediction, participants were asked to generate only one response. No confidence ratings were collected. No practice sessions were provided. At the beginning of the experiment, participants were shown the main characters (with pictures) in the TV show. The experiment was ~1 hour long and was implemented using Qualtrics.

### Video annotation

**Main experiment.** Events in the first 11 segments of the two storylines were identified by the first author (X.X.), corresponding to major plot points (total: 117; mean: 5.32 per segment; range 3–9). In addition, 74 offscreen events were identified. Of these 74 offscreen events, 43 events were identified from references in conversations during onscreen events. Another 16 events were identified based on the characters' implied movements and travels. For example, if in segment 1 character A was in place A and in segment 2 she was in place B, then the transit from place A to B for character A would be identified as an offscreen event. The remaining 15 offscreen events were identified based on logical inferences. For example, if a photograph was shown in an onscreen event (but not the act of the photograph being taken), then the action that someone took the photograph would be identified as an offscreen event. Offscreen events always occur between two contiguous segments, or before the first segment. The purpose of identifying offscreen events was to match participants' responses to video events; thus, our identification of these offscreen events was not intended to be exhaustive.

**Replication experiment.** Events in the 13 segments were identified by the authors (X.X. and X.Z.), corresponding to major plot points (total: 71; mean: 5.46 per segment; range 1–14). In addition, 66 offscreen events were identified. Of these 66 offscreen events, 47 events were identified from references in conversations during onscreen events.

Another one event was identified based on the characters' implied movements and travels. The remaining 18 offscreen events were identified based on logical inferences.

## Response analyses

Participants' retrodiction, prediction, and recall responses were minimally processed to correct obvious typos (e.g., in characters' names) and remove speculation descriptions (e.g., "I predict that..."). We discarded a small number (main experiment: $n = 20$, replication experiment: $n = 6$) of character-cued responses that did not contain references to all cued characters, along with one additional response due to the participant's misunderstanding of the task instructions during that trial in the main experiment. We carried out our analyses on the remaining 1781 retrodiction, prediction, and recall responses in the main experiment, and 878 retrodiction and prediction responses in the replication experiment.

All responses were manually coded and matched to events from the video annotations. Retrodiction and prediction responses were coded by two coders (main experiment: X.X. and Z.Z.; replication experiment: X.X. and X.Z.). Recall responses were coded by one coder (X.X.). While many responses were clearly identifiable as either matching specific storyline events or not matching any storyline events, several ambiguous cases arose. First, some responses combined or summarized over several (distinct) storyline events. Second, some responses lacked any specific detail (e.g., "character A and B talk" without describing the specific topic(s) of conversation or providing other relevant details). Based on participants' responses, in addition to the original 117 onscreen events and 74 offscreen events in the main experiment's stimulus, we added 25 new events (23 onscreen, 2 offscreen) that either summarized several events or partially matched the annotated events. In our replication study, in addition to the original 71 onscreen events and 66 offscreen events, we added 20 new events (17 onscreen, 3 offscreen). Whereas the original events were each assigned a value of one point, we assigned these additional events a half point. This point system enabled us to directly match events in participants' responses to the annotated events. In our analyses of retrodictions, predictions, and recalls, we added up the number of points earned for each response to estimate participants' event hit rates.

We coded only the first retrodiction or prediction response in each trial. For these responses, we also only considered storyline events that were in the same temporal direction as the target segment. For example, if a participant was asked to retrodict what happened in segment $n$, only events from segments 1...$n$ were considered in our analysis. When coding recall responses, we considered only events from the target segment. We also retroactively added events to the annotations that were mentioned by participants that matched events in future episodes of the TV show. We also identified and counted unmatched events in participants' responses (i.e., events that did not match any annotated events).

The television episode used in the replication study contained several interleaved storylines, which led to some deviations that were specified in our pre-registered analyses. Specifically, for our tests comparing the numbers of hits for different types of events, we additionally ran linear mixed-effect models with the type of events having two levels: matched or unmatched. This analysis was not pre-registered, but we thought that this categorization of events (matched and unmatched) would make the results clearer. For our tests of the proportions of events hit for three reference types (referenced, reference-adjacent, and remaining), we corrected the "reference-adjacent" label to correspond to individual storylines. This correction was not pre-registered.

**Resolving ambiguities and estimating inter-rater reliability.** We used Jaccard similarity to quantify the inter-rater reliabilities of the annotations, defined as the size of the intersection divided by the size of the union of the two coders' event labels for participants' responses. The Jaccard similarities were calculated for each experiment (across all trials in the uncued and cued conditions), and unmatched event labels were excluded. We observed a Jaccard similarity of 0.42 for both the main and replication experiments.

This low inter-rater reliability appeared to follow from difficulties related to setting criteria for determining whether a given response counted as a "hit" for a specific event. Whereas we had initially expected that manually matching up participants' responses with events in the narrative would be obvious, empirically, we found substantial ambiguities in this process. As one example, during one scene in our replication experiment's stimulus, the main character (Ji-Yoon) chaired a meeting for her department. One participant made a retrodiction response, "Ji-Yoon chaired a department meeting," and another participant wrote "All faculty had a meeting." If a given rater's "match" criteria included specifically mentioning that Ji-Yoon was leading the meeting, only the first participant's response would count as a "hit" for this event. However, a more lenient scorer might consider both responses to be "hits." After reviewing the scores across raters and discussing each scene on a case-by-case basis, the raters decided to re-score the responses using strict criteria (e.g., in the above example, only the first participant's response would be counted as a hit).

Another pattern we observed was that participants' guesses sometimes contained some events that actually happened (or would happen) alongside other incorrect events or details. For example, in another scene in our replication experiment's stimulus, one character (Dafna) gives another character (Bill) a ride in her car. One participant predicted that "Dafna bails Bill out and drives him back to Pembroke or helps him sober up." In one sense, if incorrect or extraneous details are ignored, this response would be considered a "hit" because the participant mentions that Dafna gives Bill a ride. However, if incorrect or extraneous details are factored into the scoring procedure (for example, Dafna never bails Bill out, nor does she help Bill sober up), the same response would be considered a miss. After reviewing the scores across raters and discussing each scene on a case-by-case basis, the raters decided to re-score the responses using the former "ignore incorrect or extraneous details" approach.

The raters repeated this general process of developing scoring criteria, comparing and discussing differences, and re-scoring the responses following those discussions until consensus was reached about every response in both experiments (i.e., Jaccard similarities of 1).

**Text embeddings of participants' responses.** To estimate the semantic similarities between pairs of responses, we first transformed each response into a 512-dimensional vector (embedding) using the Universal Sentence Encoder (Transformer USE,[16]). We defined "similarity" as the cosine of the angle formed by the responses' vectors. Following[62], we defined the "precision" of participants' responses as the median similarity between that response's vector and the embedding vectors for all other participants' recalls of the target segment (main experiment), or the similarity between that response's vector and the embedding vector for the plot synopsis of the target segment (replication experiment). We defined the "convergence" of a given response as the mean similarity between that response's vector and all other participants' responses of the same type to the corresponding segment, in the same condition. To compute these median or mean similarities, we first applied the Fisher $z$-transformation to the similarity values, then took the median or mean of the $z$-transformed similarities, and finally applied the inverse $z$-transformation to obtain the precision or convergence score.

To test the validity and reliability of the USE embeddings, we performed a classification analysis of recall responses using a leave-one-out approach. For each recall response, we calculated its semantic similarity with all other recall responses for the same storyline. We

took the segment with the highest median semantic similarity (to the recall response) as the "predicted" segment. Across all responses, the predicted segments matched the true recalled segments' labels 98.6% of the time (1088 out of 1103 predictions; chance level: 9%). We note that this validation analysis could only be carried out with data from our main experiment since we did not collect recall responses in our replication experiment.

## Reference coding

Two coders (main experiment: X.X. and Z.Z.; replication experiment: X.X. and X.Z.) identified character dialogs in the narrative that referred to past events or future (onscreen or offscreen) events. Only references to events that occurred in a different segment were included in this tagging procedure. For each reference, the source (referring) segment and the referred event number were recorded. A total of 82 references were identified in the main experiment stimulus, and 53 were identified in the replication experiment stimulus. Of these references in the main experiment, 30 referred to onscreen events, and 52 referred to offscreen events. In the replication experiment, 13 referred to onscreen events, and 40 referred to offscreen events. For these referenced events, their corresponding summary events or partial events were also labeled as referenced. In instances where the coders disagreed about a given tag, disagreements were resolved through discussions between the two coders. In our analyses, each storyline event was coded according to whether or not it had been referenced in the segment(s) that the participant had viewed thus far in the experiment.

In principle, a given event could receive multiple labels. For example, during event *A*, a character might speak about another event, *B*, during which a reference to a third event (*C*) was made. In this scenario, event *B* could be both a "referring event" ($B \to C$) and a referenced event ($A \to B$). In practice, however, this scenario was quite rare, accounting for only one out of a total of 30 onscreen events in our main experiment and one out of 13 onscreen events in our replication experiment.

## Statistical analysis

We used (generalized) linear mixed models to analyze the hit rates and numbers of events retrodicted, predicted, and recalled, as well as the precisions and convergences of participants' responses. Our models were implemented in R using the afex package. We carried out comparisons or contrasts and extracted *p*-values, using the emmeans package. Participants and stimuli (e.g., segment identity) were modeled as crossed random effects (as specified below). Random effects were selected as the maximal structure that allowed model convergence. All of our statistical tests were two-sided, and the alpha level was set to 0.05.

For our tests of the target event hit rates across four *levels* (uncued, character-cued, updated, and recall; Fig. 3B, E), we fit a generalized linear mixed model with a binomial link function:

$$\mathbf{cbind}(\text{thp}, \text{ttp} - \text{thp}) \sim \text{direction} * \text{level} * \text{seg\_cnt} * \text{storyline}$$
$$+ (\text{direction} * \text{level} \mid \text{target})$$
$$+ (\text{direction} * \text{level} * \text{seg\_cnt} \mid \text{participant})$$

where for analyses of our main experiment, *thp* was the number of points hit for the target segment, *ttp* was the total number of points for the target segment (from its annotations), *direction* was either retrodiction or prediction, *level* had four levels (uncued, character-cued, updated, and recall), *seg_cnt* represented the number of segments in the storyline that had been watched (1–10, centered), *storyline* had two levels (1 or 2), and *target* had 22 levels according to the identity of the target segment. For our analyses of our replication experiment, the *level* had two levels (uncued and character-cued), *seg_cnt* ranged from 1–12, the *storyline* parameter was omitted since there was only a single

storyline, and the *target* had 13 levels according to the identity of the target segment. In the replication experiment, we did not include random slopes of *direction* effect in the participant level in all analyses, as participants either made retrodictions or predictions (i.e., participants and tasks were nested).

For our tests of precision and convergence (Fig. 3C, D, F, and G), we fit linear mixed models using the same formula. To test the effect of *direction* (retrodiction or prediction) on target event hit rates, precision, and convergence, we fit a (generalized) linear mixed model separately for each of the three levels (uncued, character-cued, and recall).

For our tests comparing the numbers of hits for different types of events (Fig. 4B and Supplementary Fig. S6), we fit generalized linear mixed models using the same formula, but with a Poisson link function. For these models, we manually doubled the point counts to ensure that half points were mapped onto integers, ensuring compatibility with the Poisson link function.

For our analyses of the number of events hit, controlling for lag (Fig. 4C), we fit a generalized linear mixed model with a Poisson link function:

$$\text{hp\_lag} \sim \text{direction} * \text{full\_stp} * \text{lag} * \text{storyline}$$
$$+ (\text{direction} \mid \text{base\_seg}) + (1 \mid \text{base\_seg\_pair})$$
$$+ (\text{direction} * \text{full\_stp} * \text{lag} * \text{storyline} \mid \text{participant})$$

where *hp_lag* is the number of "points" earned (for each lag) in each trial (again, we manually doubled the point counts to ensure that half points were mapped onto integers, for compatibility with the Poisson link function), *full_stp* denoted whether the given events (of the given lag) were onscreen (i.e., full step) or offscreen (i.e., half step), *lag* denotes the (centered) absolute lag, *base_seg* denotes the identity of the just-watched segment (main experiment: 22 levels; replication experiment: 13 levels), and *base_seg_pair* denotes the pairing of the just-watched segment and the segment at each lag (main experiment: 440 levels; replication experiment: 324 levels).

For our analyses of the proportions of events hit for referenced versus unreferenced events (Fig. 5D, E and Supplementary Fig. S7), we fit a generalized linear model with a binomial link function:

$$\mathbf{cbind}(\text{hp\_lag}, \text{tp\_lag} - \text{hp\_lag}) \sim \text{direction} * \text{reference} * \text{full\_stp}$$
$$+ \text{lag} + (\text{direction} \mid \text{base\_seg})$$
$$+ (1 \mid \text{base\_seg\_pair})$$
$$+ (\text{direction} * \text{reference} * \text{full\_stp} + \text{lag} \mid \text{participant})$$

where *hp_lag* denotes the number of earned hit points for each reference type (referenced or unreferenced) at each lag, *tp_lag* denotes the total number of possible hit points for each reference type at each lag, and the other variables adhered to the same notation used in the above formulas.

For our tests of the proportions of events hit for all three reference types (referenced, reference-adjacent, and remaining: Fig. 6D, E and Supplementary Figs. S9, S10; or referenced, referring, and other: Fig. 7D and Supplementary Fig. S11), we fit a generalized linear mixed model using the same formula as above, but with three (rather than two) *reference* levels.

Several of our analyses entailed comparing the relative hit rates or probabilities of two different conditions or outcomes. We used the emmeans package to compute the odds ratios given the generalized linear mixed models we fit for the given analysis. These odds ratios reflect the odds (calculated as $\frac{p}{1-p}$), where *p* is the probability that the outcome occurs) of a particular outcome (e.g., making a response about a particular event) given a scenario (e.g., the event occurred prior to the just-watched segment) divided by the odds of the outcome occurring in the alternative scenario (e.g., the event occurred after the just-watched segment).

## Large-scale analysis of conversational data

At a high level, the goal of our analysis of conversational data was to predict in-text references to past and future events. Manually identifying these references is labor and time-intensive, so it is impractical to scale up manual tagging to millions of documents. Instead, we defined a set of heuristics for predicting when text is referring to real or hypothetical past or future events. Our approach comprised four main steps.

First, we used the nltk package[63] to segment each document into individual sentences. Each sentence was processed independently of the others. Second, we handled contractions using the contractions package (e.g., "we'll" was split into "we will," and so on). Third, we defined two sets of "keywords" (words and phrases) that tended to be indicative of referring to the past (Tab. S6) or future (Tab. S7). We used ChatGPT[64] to generate each list, with exactly 50 templates per list, using the following prompt:

*I'm designing a heuristic algorithm for identifying references (in text) to past and future events. Part of the algorithm will involve looking for specific keywords or phrases that suggest that the text is referring to something that happened (or will happen) in the past and/or future. Could you help me generate a list of 50 keywords or phrases to include in each list (one list for identifying references to the past and a second list for identifying references to the future)? I'd like to be able to paste the lists you generate into two plain text documents with one row per keyword or phrase, and no other content. Please output the lists as a "code" block (enclosed by "'...'").*

Fourth, we used part-of-speech tagging (again, using the nltk package) to look for verbs or verb phrases that were in past or future tenses. After the words were tagged with their predicted parts of speech, we used regular expressions (applied to the sequences of tags) to label each verb or verb phrase with a human-readable verb form (e.g., "future perfect continuous passive," "conditional perfect continuous passive," and so on). The regular expressions we used to generate these labels are shown in the Supplementary Table S4, and the part of speech tags are defined in the Supplementary Table S5.

We treated each keyword match (of past or future keywords) as a "reference" (to a past or future event, respectively). We also tracked whether any past or future verb forms were detected. We then tallied up the numbers of past and/or future references across sentences within the given document, counting (up to) one past reference and one future reference per sentence.

In designing the above approach, we used the transcript of "The Chair", Episode 1 to "debug" the automatically derived tense tags. We began by spot-checking randomly selected sentences from the episode's transcript, tweaking the algorithm as needed to catch tricky edge cases. Once we began to see good performance on the excerpted sentences, we applied the approach to the full episode transcript. We verified that the automated procedure accurately recovered the approximate relative numbers of past versus future "references" that we identified by hand. Next, we applied the automated tagging procedure to the other episodes of "The Chair". We found that our approach appeared to generalize to those episodes as well, as compared with manually derived labels. Finally, after computing automated tags for the other datasets, we carried out a final set of "spot checks" on randomly excerpted utterances from each dataset to verify that the automated tags were behaving as expected.

In general, our automated tagging procedure tended to overcount the number of references (Supplementary Fig. S12A). From manually examining hundreds of example tags, we noticed that our automated tagging procedure often counts the "same" references

multiple times when the reference is extended across multiple sentences. Specifically, the manually generated tags sought to identify references to specific events that occurred or were implied to occur in other parts of the narrative. In contrast, as a heuristic, we designed the automated tagging procedure to identify uses of the past or future tense as a proxy for references to past or future events. Individual conversations often contain multiple references to a given (past or future) event. Whereas the manually generated tags counted these as "single" references, our automated tagging procedure had no means of differentiating between several references to the same event versus the same number of references to different events. This leads the automated tagging procedure to overestimate the number of distinct events being referenced. These overestimates tended to be biased towards future references (Supplementary Fig. S12 B, C). Therefore, the ratios of past to future references we estimated automatically appeared to systematically underestimate the true ratio. Nevertheless, underestimating the Past/Future reference ratio does not change our conclusion that past references are more common than future references in human-human conversations.

## Reporting summary

Further information on research design is available in the Nature Portfolio Reporting Summary linked to this article.

## Data availability

Behavioral data generated for this study have been deposited on Zenodo (https://doi.org/10.5281/zenodo.12522455)[65]. Data included in our large-scale analysis are available at https://imsdb.com, https://convokit.cornell.edu/documentation/datasets.html, https://scrapsfromtheloft.com/?s=THE+CHAIR, https://github.com/ricsinaruto/gutenberg-dialog, and https://www.screenspy.com/the-chair-season-1-episode-1/, and on Zenodo (https://doi.org/10.5281/zenodo.12522455)[65].

## Code availability

All of the analysis code from our paper is available on Zenodo (https://doi.org/10.5281/zenodo.12522455)[65].

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

## Acknowledgements

We thank Luke Chang, Yi Fang, Paxton Fitzpatrick, Caroline Lee, Meghan Meyer, Lucy Owen, and Kirsten Ziman for feedback and scientific discussions. Our work was supported in part by NSF CAREER Award Number 2145172 to J.R.M. The content is solely the responsibility of the authors and does not necessarily represent the official views of our supporting organizations. The funders had no role in study design, data collection and analysis, decision to publish, or preparation of the manuscript.

## Author contributions

Conceptualization: X.X. and J.R.M.; Methodology: X.X. and J.R.M.; Software: X.X. and J.R.M.; Analysis: X.X., Z.Z., X.Z., and J.R.M.; Writing, Reviewing, and Editing: X.X., Z.Z., X.Z., and J.R.M.; Supervision: J.R.M.

## Competing interests

The authors declare no competing interests.
