## [Peer Review File · Nature Communications]

The psychological arrow of time drives temporal asymmetries in inferring unobserved past and future eventsREVIEWER COMMENTS

Reviewer #1 (Remarks to the Author):

The authors present data from one laboratory experiment assessing the capacity for participants to predict what might have happened before or what might happen after a given segment a television show storyline. Among other things, the authors report that participants were better at predicting what might have occurred leading up to a segment than what might occur after a segment and suggest that the tendency for past information to be mentioned in conversation and its proximity to the present may help to account for this pattern of data. The authors argue that these findings may help us to better understand how we understand other people.

This is a well-written paper that uses a sophisticated experimental paradigm to study prediction about the past and future under controlled settings. While I believe the data are interesting, my primary concern is that it is not clear whether the data are generalizable beyond the very specific context that is tested in this study. First, the conclusions that the authors draw are based on a single experiment with a relatively small sample (for which no justification is given). Second, the authors use story lines from one television show, and it is not clear whether these findings generalize to other experimental stimuli. Finally, the rationale as to how this paradigm might inform real world interactions is somewhat weak. The authors provide examples of meeting someone for the very first time and watching a movie starting somewhere at its midpoint. These situations are rare compared to the many different circumstances in which people are constantly making predictions about others in terms of their past and/or future (e.g., thinking about the past and future of familiar others). The study also fails to consider the possibility that people may simply refrain from making predictions about others if they do not have any relevant information available to them.

Again, I believe this is a very clever paradigm and the data are interesting, but it is not clear that they tell us much about how we make predictions about others in the real world. Moreover, the reported findings simply need to be replicated and extended across different participant samples and experimental stimuli to convince the reader as to their reliability.

Minor points:

Whereas the authors very clearly explained the procedure in the body of the manuscript, I found Fig. 2 somewhat confusing.

The methods are repeated in some detail in the results section. This is fine, but the initial part of the results and the methods are rather repetitive.

The authors do not report interrater reliability where coding included two coders.

Reviewer #2 (Remarks to the Author):

In this paper, Xu, Zhu, and Manning report on a study examining temporal asymmetries when inferring unobserved events in the past and the future. In the study, participants watched segments of a television series (*Why Women Kill*), and performed several task conditions alternating between viewing, recall, and either retrodiction or prediction of an unviewed event. Additional task conditions divided trials into uncued and character-cued prediction or retrodiction. A final condition was an "updated prediction" in which participants attempted to guess what happened in a given event having been provided both the preceding event (which was used to predict initially) and following event (which presumably led to a clearer prediction of the unseen middle event). The study produced several findings. First, hit rate (i.e., proportion of correctly retrodicted, predicted, or recalled events) were better for character-cued than uncued retrodiction or prediction, and more importantly, hit rates were higher and transcripts more precise for retrodiction than prediction. Second, participants showed a greater propensity to retrodict to distant (i.e., lag greater than 1 event away from the cued event) past events than to predict to distant future events. Third, the stimulus itself seemed to drive these effects to some extent. That is, the characters in the show made more reference to past events than future events, which seems to have driven the bias toward retrodiction to some extent in the behavioral data. Fourth, events near these references by characters in the show seemed to drive higher hit rates. Fifth, this hit rate applied to the event being referenced, but not the referring event (i.e., the event from

which the reference originated rather than the one targeted by the reference).

Overall, I found this to be an interesting paper targeting a very interesting topic. Given the association between episodic memory and future thinking, it is interesting to see an investigation into the potential biases in directionality that may mediate such an apparently shared resource. Nonetheless, I have a number of issues with the paper in its current form, which I list below (in no particular order):

1. The introduction is very light on discussion of highly relevant work in the field on memory in both basic and more naturalistic experiments, particularly temporal influences on memory, and temporal biases in memory. Consequently, I think that there is a rather weak representation of relevant works being cited here. I found the introduction to be entertaining to read, to be clear, but it felt more based on speculation and commonsense notions of how things should or might work than built on a foundation of empirical evidence that led to specific questions. I think this could stand to be sharpened

2. Relatedly, I did not feel that the analyses in the paper built on one another in a satisfying way. The initial analyses in Figure 3 made a lot of sense given the introductory lead-in and the motivation for designing the experiment, and the data described in Figure 4 seemed to logically follow from that initial set of comparisons. But after this point, things stopped flowing for me. There did not seem to be a lot of logic stringing these analyses together. Rather, it felt as if the authors collated a set of interesting analyses without really guiding the reader through why that analysis was important, how it built on the prior findings, or why it really needed to happen in the grander scheme of things. Though this is only one example, on Line 297, the authors write: "If there are associations and dependencies between temporally adjacent events, might characters' references to specific events also boost participants' estimates of other events that were temporally adjacent to the referenced events?" At the time of reading, I completely missed the logic underlying the move toward this prediction. (As an aside, this is also a bit convoluted a sentence.) To be clear, the findings are interesting and make better sense in the larger scope of the paper once I had time to step back and digest it all. However, in my initial read-through, I lost focus of the overall "point" and rather felt like I was being led down a garden path of exploratory analyses. I have to recommend tightening up the logic throughout the paper.

3. At the top of Page 5, the authors cite Radvansky and Copeland, Zwaan and Radvansky, Bower et al., and Ranganath and Ritchey's work in reference inferences about other people's lives in terms of event schemas, scripts, or situation models. They assert that "the accuracy of inferences about the past and the future of others' lives should be approximately equal" on the basis of these cited works. I actually am not sure if the cited authors would make any claims about the directionality of inferences into the past or future, or a lack thereof. In fact, I actually doubt there would be any such claim of an "approximately equal" distribution on the part of several of these individuals.

4. I am a bit unclear on the tagging of implicit events in half-step lags, and exactly how the specifics of this choice were implemented. Was it determined that offscreen events must have come between onscreen events? That is, if a cued event has a target segment of lag -2, and there is an intervening segment of lag -1, did the experimenters decide that some event that must have taken place between -2 and -1 must be lag -1.5? This seems reasonable, but it is not explained clearly. However, if this is the case, it does raise the question of how confidently the experimenters could determine between which of two specific events an offscreen event falls. I would like to see more clarification on this.

5. I did not find that Figure 5, Panel E was explained very clearly. Perhaps more importantly, though, it is quite difficult to see the key data (i.e., the colored sliver at the far left of the gray bar) in these plots. I am not really sure what to recommend to the authors that would "improve" the plot, but it really is not a great viewing experience for the reader.

6. I am assuming that Figure 8 is comprised of data from the prior analyses just collated into a plot, but this isn't clear. If it is hypothetical, this should be specified.

7. A larger issue I have with these data is that I am led to question the generalizability of the bias toward retrodiction due to the stimulus itself. One of the major analyses in this paper shows that the stimulus itself (i.e., references made by the characters) directs the viewer more toward the past than toward the future (Fig. 5). Further, these references seem to directly influence

participants' behavior in the experiment (Fig. 6). How, then, can we be at all confident that the "psychological arrow of time" directs us toward the past in a general sense, and not simply as a matter of cueing in this particular experiment? The results of this study seem pretty unsurprising given the aforementioned bias toward the stimulus itself referencing the past, which leads me to wonder why we would expect any other result here. At the very least, I think that this is a major enough complication with generalizability that the broader statements made throughout the paper such as "an underlying knowledge asymmetry in favor of the past" should perhaps be dialed back a bit. To be fair, the authors lay out a nice logical story for why past knowledge should exceed future knowledge in the introduction. However, there are in my view pretty major complications with the stimulus itself that limit one's ability to make strong statements on the back of the data, and perhaps even lead to questions about the impact or importance of the findings in the absence of a more temporally-agnostic stimulus.

8. Although the discussion is certainly the place for speculation, it felt very speculative. In particular, there were mentions of the "thermodynamic" arrow of time and "detective and forensic science" which, while interesting, are not as relevant as psychological and neuroscientific phenomena that could be discussed here. In line with my first comment about the introduction, this section was an interesting read, but it unfortunately felt shallow in terms of relating to and citing relevant literature.

9. A very minor nitpick, but it would be nice to include a quick rundown of odds ratios, at least in the methods section. This is certainly an appropriate way of analyzing these data, but I find it likely that a number of prospective readers might not immediately grasp what was being evaluated here.

Reviewer #3 (Remarks to the Author):

Since the time of Ebbinghaus, memory research has largely focused on carefully controlled lists of stimuli. One of the important phenomena this work has revealed is the contiguity effect, the finding that, all else equal, memory for an event brings to mind memory for nearby stimuli. Over roughly the last decade or so, there has been a movement to study human memory under more naturalistic circumstances. Much of this work has relied on participants experiencing narratives, such as radio shows or oral story-telling or, as in this study, television programs. Because the study materials are so different, it is difficult to assess the source of differences between laboratory memory studies and studies using narrative material. After all, narratives often have a rich structure with layers of semantic meanings and emotional and cultural shadings. It is difficult to measure these variables from narratives and thus difficult to understand their separate and cooperative effects. The present study presents an ingenious method to estimate the connectedness of the narrative. One might say that this study measures the contiguity effect in the complete absence of memory for the study materials.

Participants are presented with brief clips from an unfamiliar television program and asked to predict events that will follow the clip and retrodict events that preceded the clip. In this way it is possible to trace out something analogous to a contiguity effect that is solely attributable to the participants' expectations about the statistical structure of the narrative. The primary result is that, whereas the contiguity effect in laboratory memory experiments is typically asymmetric favoring memory in the forward direction, the ability to predict unobserved events from a narrative is asymmetric in the backward direction. On its face, this result is not only inconsistent with the contiguity effect from laboratory memory experiments, but also from laboratory experiments that directly evaluate participants' abilities to estimate the past and the future from Markov sequences of stimuli. Jones & Pashler (2007) observed no evidence for an asymmetry, in either direction, when participants were asked to judge the stimulus that followed or preceded a cue stimulus.

So what accounts for the counterintuitive result in the present study. The authors argue that the fact that in the narrative, characters talk about past events more frequently than future events. This at least partially accounts for the backward asymmetry, which seems to at least be reduced for events that were not referenced. Moreover, the authors argue that references enhance the ability to predict events surrounding the referenced events. For instance, if a character at work mentions that they shoveled their driveway that morning, one might guess that it snowed in the night and that they drove to work.

In summary, this paper is methodologically innovative and tackles a problem that is of importance

in the study of human memory. The analyses are careful and sound. Like many studies that use cultural artifacts as study materials, it is unclear how well this generalizes to actual real world experiences, or even other cultural artifacts. Narratives are crafted to be unpredictable; a television show that is too predictable is boring. However, for the narrative in this study at least, it seems that asymmetry in the ability to infer events in time was at least to some extent attributable to the memory conveyed by the characters in the program.

One might ask if temporal inference of events is asymmetric for narratives that do not explicitly convey information about the past or the future (assuming the Jones & Pashler results are the last word on genuinely Markov processes). Even those without characters describing past events. Simple statistical structures may yield asymmetric predictions from a normative model. For instance, suppose that a stimulus Y happens with relatively high probability according to a Poisson process. The stimulus train is assembled such that every once in a while, Y is preceded by X at precisely 1 second. That is $P(Y \text{ presented } 1 \text{ s in the future} \mid X \text{ presented now}) = 1$, whereas $P(X \text{ was presented } 1 \text{ s in the past} \mid Y \text{ presented now})$ is very small. Similarly, taking into account the imprecision of temporal memory predictions that can be formed from the sequence X Y Z are asymmetric. Starting from Y, the time of X is more uncertain than the time of Z, simply because of our ability to resolve temporal intervals.

Reviewer #4 (Remarks to the Author):

The authors report evidence that people are better able to retrodict other people's pasts than predict other people's future. Past work has shown similar effects for people's own personal experience, but this is the first work to show that the effects extend to how we understand other people and their experiences. The authors further showed that the temporal asymmetry in character retrodictions/predictions was related to (i) characters' tendencies to refer to past events more than future events in their ongoing conversations, and (ii) associations between temporally proximal events. There is a lot to like about this paper: the results are compelling, the experimental design is creative, and it is a novel finding. There are a few concerns that if addressed in a revision would strengthen the manuscript and make it better suited for publication.

1. Television writers presumably spend a good chunk of time trying to write scripts that viewers cannot easily predict. When as a viewer you can easily predict what happens next in a show, it's less entertaining--people frequently complain about shows that are "too predictable." Simultaneously, tv writers want to make the story easy to remember so people can jump in and out, or remember what happened in the last episode so they can enjoy the next episode a week later. This all makes me wonder if the temporal asymmetries observed here generalize to other scenarios that are not scripted and not-predesigned to be more or less predictable vs. retrodictable. If the authors were able to replicate their results with footage of people in unscripted, everyday conversations, that would be very compelling. Or at the very least, this would be good to address as a limitation in the discussion section.

2. Is it possible to dive even deeper into the features of the story that are better retrodicted than predicted? Is it interpersonal events that are best retrodicted? Characters' internal thoughts/feelings/beliefs? Key narrative transition points? Something else? The authors note that it is partly what characters say about the past, but it would be interesting if the effect is driven by certain content communicated.

3. A replication would also allow the authors to potentially test some questions about the psychological processes driving the effect. In social cognition research, a common finding is that people "use the self" to simulate other people (see some relevant citations below). And, they use this strategy more for people they believe are similar to themselves. Given that your findings parallel how people's retrodictions/predictions about themselves work, it seems possible that this "simulation from self" strategy is at work here (particularly if analyses related to the question above suggest retrodiction/prediction for "invisible mental states" correspond with this temporal asymmetry). If a large online study was run (either with the same stimuli reported here or different stimuli that's unscripted), you could also ask participants how similar they feel to each character and test whether that modulates the results. Alternatively, if the authors do not want to run such a study, the simulation from self strategy could be proposed or discussed in the

discussion.

Tamir, D. I., & Mitchell, J. P. (2013). Anchoring and adjustment during social inferences. *Journal of Experimental Psychology: General*, 142(1), 151.

Meyer, M. L., Zhao, Z., & Tamir, D. I. (2019). Simulating other people changes the self. *Journal of Experimental Psychology: General*, 148(11), 1898.

Reviewer #1 (Remarks to the Author):

The authors present data from one laboratory experiment assessing the capacity for participants to predict what might have happened before or what might happen after a given segment a television show storyline. Among other things, the authors report that participants were better at predicting what might have occurred leading up to a segment than what might occur after a segment and suggest that the tendency for past information to be mentioned in conversation and its proximity to the present may help to account for this pattern of data. The authors argue that these findings may help us to better understand how we understand other people.

We generally agree with the reviewer's summary here. However, we do wish to clarify that our major claim is not that mentions of past or future events help us better understand other people, but rather we claim that these references help us to gain insights into other times (beyond "now"). There is some overlap between what we mean by other "times" versus other "people," since in our setup the most interesting insights are gained about other people's (e.g., characters') experiences (at moments before or after a just-watched video segment).

This is a well-written paper that uses a sophisticated experimental paradigm to study prediction about the past and future under controlled settings. While I believe the data are interesting, my primary concern is that it is not clear whether the data are generalizable beyond the very specific context that is tested in this study. First, the conclusions that the authors draw are based on a single experiment with a relatively small sample (for which no justification is given). Second, the authors use story lines from one television show, and it is not clear whether these findings generalize to other experimental stimuli. Finally, the rationale as to how this paradigm might inform real world interactions is somewhat weak. The authors provide examples of meeting someone for the very first time and watching a movie starting somewhere at its midpoint. These situations are rare compared to the many different circumstances in which people are constantly making predictions about others in terms of their past and/or future (e.g., thinking about the past and future of familiar others). The study also fails to consider the possibility that people may simply refrain from making predictions about others if they do not have any relevant information available to them.

Again, I believe this is a very clever paradigm and the data are interesting, but it is not clear that they tell us much about how we make predictions about others in the real world. Moreover, the reported findings simply need to be replicated and extended across different participant samples and experimental stimuli to convince the reader as to their reliability.

In our original study, we reported three core findings:

1. Participants more readily retrodict unknown past events than they predict unknown future events.
2. Characters in the narrative (television episode) we examined tended to talk about events from the past more often than events from the future.
3. Participants' retrodictions and predictions of unknown parts of the television episode tended to follow from what the characters talked about (e.g., specific things they referenced or events nearby in time to things the characters referenced).

The reviewer raises some important concerns here. First, since our study comprised a single experiment and a relatively small sample size, to what extent might our findings extend to other settings or people? Second, how generalizable are our findings to other situations or stimuli? And third, what does our work tell us about how we make retrodictions and predictions in the real world?

We have approached these questions using two new experiments. In our first new experiment, we carried out a pre-registered replication study using a new stimulus and participant pool. We replicated our major findings, as detailed in our revised results section (pages 5–24) and in our supplemental materials. Our major takeaway of the replication study is that participants' behaviors that we observed in our original study are unlikely to be due to some quirk of the particular stimulus or participant pool, since we found similar behaviors with a new stimulus and participant pool.

We also considered the possibility that perhaps our findings might reflect some property of television shows, movies, etc., more generally, in ways that are not actually representative of real-world conversations. For example, perhaps characters in television shows (or movies, novels, and other stimuli crafted for entertainment purposes) are written to talk about the past so as to make the narrative more unpredictable and therefore more engaging. Or perhaps something else about the writing process leads authors to bias their written dialogues in favor of the past. Or, even if dialogue in television shows and/or movies *is* reflective of real-world experiences and conversations on average, it could be that the particular shows we selected are not representative along some other important dimension.

To examine these latter possibilities, we carried out a meta analysis on several datasets, collectively comprising tens of millions of conversations and nearly half a billion words. The datasets covered a range of media including transcripts from television shows and movies, transcripts of natural spoken and written conversations, and conversations excerpted from novels. We used a natural language processing approach (pages 40–42) to identify references in each document to past and/or future events. We then tallied up the numbers of

references. Overall, as shown in Figure 8 of our revised manuscript, we found that references to past events are 1.45 times more prevalent than references to future events, regardless of the specific types of conversation we examined (written or spoken dialogues, scripted or unscripted, etc.). We suggest that a bias towards referencing the past (versus the future) is a common tendency of human conversation in general, although of course this does not mean that *every* individual conversation shows this bias.

Finally, the reviewer brings up some important concerns about how realistic or meaningful our general setup might be— e.g., whether it is reflective of something that happens often in everyday life. We wish to clarify that we are *not* trying to suggest that the examples of “starting a movie part way through” or “predicting what happened in the past in a stranger’s life” (and so on) are something we are likely to encounter in the normal course of everyday life. Nor do we attempt to make claims about how often we make these sorts of inferences in our everyday lives (in our study, we effectively “forced” participants to retrodict the past or predict the future). Rather, our intention was to attempt to separate out two aspects of our experience that are typically conflated: (a) our *experience* of the past versus the future, and (b) our ability to make *inferences* about the past versus the future. As we explain in the introduction (e.g., Figure 1), in our own lives we nearly always know more about the past (since we experienced it, and since we often remember our experiences). This means that, if we were to ask participants to make “guesses” about their own past versus future experiences, they’d almost certainly be better at guessing about the past. But going *beyond* our own experiences (e.g., where we *can’t* rely on our own specific memories to drive our insights), is there anything about the past itself (versus the future) that provides an information asymmetry to us in the present moment? That’s what our studies are focused on, and that is why we designed an experimental paradigm that enabled us to distinguish between what the *participant* experienced in the past (or what they would experience in the future) and what happened in the past or future of the *narratives*, relative to the current moment.

Following intuitions from prior studies of statistical learning (e.g., learning to retrodict or predict sequences), going into our study we wondered whether the past and future are “symmetric” (i.e., whether the present tells us about as much about each) or whether there might be some other aspect of everyday experiences in the “real” world that perhaps differed from the sorts of random or first-order Markov process sequences that are typically used in these sequence learning studies. We ended up doing a deep dive into several other literatures, on statistical physics, philosophy of time, and time perception. Those literatures are quite complicated, and a lot of the insights we gained go beyond the scope of the current manuscript. Therefore we have chosen instead to focus in on one particular aspect of this

temporal asymmetry question. We were surprised (but delighted!) to make some real discoveries in our study that could help to advance this area of inquiry. Essentially, we found that the so-called “psychological arrow of time” that gives us asymmetric knowledge of our past (due to our memories) can be “communicated” to other people through conversations.

Minor points:

Whereas the authors very clearly explained the procedure in the body of the manuscript, I found Fig. 2 somewhat confusing.

We have added some clarifying text to the Figure 2 caption.

The methods are repeated in some detail in the results section. This is fine, but the initial part of the results and the methods are rather repetitive.

We appreciate the reviewer’s point here. Because we recognize that our experimental paradigm is unusually complicated, we chose to repeat some of the most important descriptions (where relevant) in the results section rather than require readers to “hold the full experiment in mind” as they parse our results. We experimented with several alternative formats following the reviewer’s comment, but we feel that the current presentation optimizes clarity (at the expense of some repetition, as the reviewer notes). At several reviewers’ (and the editor’s) request, our revised manuscript also incorporates a second “replication” experiment along with a new meta analysis, which we worried might place an even greater “working memory load” on readers were we to relegate all details solely to the methods section.

The authors do not report interrater reliability where coding included two coders.

We have added a section reporting inter-rater reliability and describing how labeling differences across raters were resolved prior to carrying out our analyses of the response data (pages 35–36):

“We used Jaccard similarity to quantify the inter-rater reliabilities of the annotations, defined as the size of the intersection divided by the size of the union of the two coders’ event labels for participants’ responses. The Jaccard similarities were calculated for each experiment (across all trials in the uncued and cued conditions), and unmatched event labels were excluded. We observed a Jaccard similarity of 0.42 for both the main and replication experiments.

This low inter-rater reliability appeared to follow from difficulties related to setting criteria for determining whether a response counts as a “hit” for a specific event. Whereas we had initially expected that manually matching up participants' responses with events in the narrative would be obvious, empirically we found substantial ambiguities in this process. As one example, during one scene in our replication experiment's stimulus, the main character (Ji-Yoon) chaired a meeting for her department. One participant made a retrodiction response “Ji-Yoon chaired a department meeting” and another participant wrote “All faculty had a meeting.” If a given rater's “match” criteria included specifically mentioning that *Ji-Yoon* was *leading* the meeting, only the first participant's response would count as a “hit” for this event. However, a more lenient scorer might consider both responses to be “hits.” After reviewing the scores across raters and discussing each scene on a case-by-case basis, the raters decided to re-score the responses using strict criteria (e.g., in the above example, only the first participant's response would be counted as a hit).

Another pattern we observed was that participants' guesses sometimes contained some events that actually happened (or would happen) alongside other incorrect events or details. For example, in another scene in our replication experiment's stimulus, one character (Dafna) gives another character (Bill) a ride in her car. One participant predicted that “Dafna bails Bill out and drives him back to Pembroke or helps him sober up.” In one sense, if incorrect or extraneous details are ignored, this response would be considered a “hit” because the participant mentions that Dafna gives Bill a ride. However, if incorrect or extraneous details are factored into the scoring procedure (for example, Dafna never bails Bill out, nor does she help Bill sober up), the same response would be considered a miss. After reviewing the scores across raters and discussing each scene on a case-by-case basis, the raters decided to re-score the responses using the “ignore incorrect or extraneous details” approach.

The raters repeated this general process of developing scoring criteria, comparing and discussing differences, and re-scoring the responses following those discussions until consensus was reached about every response in both experiments (i.e., Jaccard similarities of 1).”

Reviewer #2 (Remarks to the Author):

In this paper, Xu, Zhu, and Manning report on a study examining temporal asymmetries when inferring unobserved events in the past and the future. In the study, participants watched segments of a television

series (*Why Women Kill*), and performed several task conditions alternating between viewing, recall, and either retrodiction or prediction of an unviewed event. Additional task conditions divided trials into uncued and character-cued prediction or retrodiction. A final condition was an “updated prediction” in which participants attempted to guess what happened in a given event having been provided both the preceding event (which was used to predict initially) and following event (which presumably led to a clearer prediction of the unseen middle event). The study produced several findings. First, hit rate (i.e., proportion of correctly retrodicted, predicted, or recalled events) were better for character-cued than uncued retrodiction or prediction, and more importantly, hit rates were higher and transcripts more precise for retrodiction than prediction. Second, participants showed a greater propensity to retrodict to distant (i.e., lag greater than 1 event away from the cued event) past events than to predict to distant future events. Third, the stimulus itself seemed to drive these effects to some extent. That is, the characters in the show made more reference to past events than future events, which seems to have driven the bias toward retrodiction to some extent in the behavioral data. Fourth, events near these references by characters in the show seemed to drive higher hit rates. Fifth, this hit rate applied to the event being referenced, but not the referring event (i.e., the event from which the reference originated rather than the one targeted by the reference).

Overall, I found this to be an interesting paper targeting a very interesting topic. Given the association between episodic memory and future thinking, it is interesting to see an investigation into the potential biases in directionality that may mediate such an apparently shared resource. Nonetheless, I have a number of issues with the paper in its current form, which I list below (in no particular order):

1. The introduction is very light on discussion of highly relevant work in the field on memory in both basic and more naturalistic experiments, particularly temporal influences on memory, and temporal biases in memory. Consequently, I think that there is a rather weak representation of relevant works being cited here. I found the introduction to be entertaining to read, to be clear, but it felt more based on speculation and commonsense notions of how things should or might work than built on a foundation of empirical evidence that led to specific questions. I think this could stand to be sharpened

To help flesh out our review of the existing literature, we have added an additional discussion (e.g., pages 25, 27-28) of prior work on time perception, along with studies that examine how people think about and communicate about the past, present, and future, and how they “move through time” in their memories of their prior experiences, or in their imaginings of past or future events.

That said, we sympathize with the reviewer’s point that several parts of the introduction and discussion are based on speculation and “commonsense notions.” A challenge we faced is that a number of aspects of our work, to our knowledge at least, have not been formally

studied or tested in prior work— rather, we suspect they have been chalked up to “commonsense notions” in most other studies. For example, most studies of episodic memory assume that people remember their past experiences but not their future experiences, that time moves (for everyone) from the past to the future, and so on. We often take these notions for granted. One of the points we tried to make in our paper, however, is that there might be some intellectual benefit to thinking through those sorts of assumptions (using thought experiments, common sense examples, and so on, in addition to drawing on prior work from several fields and running new experiments). We think it brings up some deep questions, many of which are beyond the scope of our current paper, about the fundamental nature of memory, how we think, how we perceive time, physical laws of the universe (with respect to time), and so on. We touch on some of these ideas in the introduction and discussion, although we also tried not to stray too far from our main focus of reporting our empirical findings.

2. Relatedly, I did not feel that the analyses in the paper built on one another in a satisfying way. The initial analyses in Figure 3 made a lot of sense given the introductory lead-in and the motivation for designing the experiment, and the data described in Figure 4 seemed to logically follow from that initial set of comparisons. But after this point, things stopped flowing for me. There did not seem to be a lot of logic stringing these analyses together. Rather, it felt as if the authors collated a set of interesting analyses without really guiding the reader through why that analysis was important, how it built on the prior findings, or why it really needed to happen in the grander scheme of things. Though this is only one example, on Line 297, the authors write: “If there are associations and dependencies between temporally adjacent events, might characters’ references to specific events also boost participants’ estimates of other events that were temporally adjacent to the referenced events?” At the time of reading, I completely missed the logic underlying the move toward this prediction. (As an aside, this is also a bit convoluted a sentence.) To be clear, the findings are interesting and make better sense in the larger scope of the paper once I had time to step back and digest it all. However, in my initial read-through, I lost focus of the overall “point” and rather felt like I was being led down a garden path of exploratory analyses. I have to recommend tightening up the logic throughout the paper.

We appreciate this point. We have added some additional text throughout the results section to unpack the underlying logic a bit more, including the transition the reviewer calls out here (page 17).

In general, we have organized the results section using the following logic, which we hope is clearer in our revised manuscript:

- Our overarching question is: given that neither the past and future have been directly observed, are participants better at guessing about the past, better at guessing about the future, or about equally good at guessing about the past and future?
- Figure 3: Sanity checks on performance metrics– participants’ responses are better (via several metrics) when they are provided with more information (uncued guesses < cued guesses < recall).
- Figure 4: Participants are generally better at retrodicting the unobserved past than predicting the unobserved future. We observe this bias, not just for events immediately before or after the just-watched movie segment, but also for events from more temporally distant segments.
- Figure 5: Where do these biases come from? Is it something inherent to the participants, or does it come from the stimuli? When we dug into the content of the stimuli, we found that the characters in the narratives tended to talk about the past more than the future. Participants’ responses tended to follow from what the characters spoke about. This tells us that at least some of participants’ behaviors can be explained from the stimuli.
- Figure 6: Can participants’ behaviors be explained solely by what characters in the television shows said, or are there other factors at play as well? We found that, in addition to seeing a “boost” for (past or future) events that characters specifically mention, we also see higher hit rates for events that are temporally *near* those mentioned (“referenced”) events. We think this is due to associations between temporally proximal events that participants can infer. (As mentioned above, we’ve added some new text to unpack this logic; page 17.)
- Figure 7: When a character mentions a past or future event during segment *n*, we know (from Figs. 5 and 6) those referenced events get a boost in hit rate, along with other events that happened nearby in time. But does segment *n* also receive an inference benefit for having been the “source” of participants’ guesses? We see no evidence to support this. So conversations are asymmetric: talking about an event (in the past or future) boosts the listener’s ability to infer what happened at the referenced times, but at the moment being referenced there is no indication that listeners can intuit that the given moment will (or has been) mentioned by the characters in the future or past.
- Figure 8 (new): Even though characters in *this* television show (and, in the updated manuscript, in our replication study’s television show) happen to talk more about the past than the future, how generally does this hold? Is it merely a property of these specific shows? Or of television shows or movies in general? Or most fictional or written work? We ran a meta analysis of tens of millions of conversations from a variety of sources (video transcripts from television shows and films, novels, and

written and spoken natural conversations) and found that references to past events occur 1.45 times more often than references to future events. Therefore it seems that a bias in favor of the past is a general property of many fictional and real human conversations.

- Figure 9 (in our previous draft, this was Figure 8): A “cartoon” summarizing our main claims regarding how much can be inferred about the past and future by observing the present.

3. At the top of Page 5, the authors cite Radvansky and Copeland, Zwaan and Radvansky, Bower et al., and Ranganath and Ritchey’s work in reference inferences about other people’s lives in terms of event schemas, scripts, or situation models. They assert that “the accuracy of inferences about the past and the future of others’ lives should be approximately equal” on the basis of these cited works. I actually am not sure if the cited authors would make any claims about the directionality of inferences into the past or future, or a lack thereof. In fact, I actually doubt there would be any such claim of an “approximately equal” distribution on the part of several of these individuals.

We have added a clarification that the claim of “symmetry” comes from our own intuitions as opposed to from those authors (pages 4-5):

“We note that the aforementioned authors make no specific claims about temporal symmetries or asymmetries. Rather, we claim that statistical regularities might *imply* symmetry (e.g., if you are on step n of an unfolding schema, this suggests you may have just completed step $n - 1$ and that you may next encounter step $n + 1$).”

4. I am a bit unclear on the tagging of implicit events in half-step lags, and exactly how the specifics of this choice were implemented. Was it determined that offscreen events must have come between onscreen events? That is, if a cued event has a target segment of lag -2, and there is an intervening segment of lag -1, did the experimenters decide that some event that must have taken place between -2 and -1 must be lag -1.5? This seems reasonable, but it is not explained clearly. However, if this is the case, it does raise the question of how confidently the experimenters could determine between which of two specific events an offscreen event falls. I would like to see more clarification on this.

The reviewer’s description is correct– if an event occurred “between” segments n and $n + 1$, we labeled it as having occurred during segment “ $n + 0.5$ ”. With respect to the “lag” analysis, here’s how we summarize the half step lag assignments in the results section (pages 12):

“We tagged offscreen events using half steps. For example, an offscreen event that occurred after the prior segment but before the just-watched segment

would be assigned a lag of -0.5.”

We of course cannot be 100% confident when an event that was never actually depicted “occured” in the narrative. That said, in most cases the timings were straightforward. For example (page 11):

“...a character in location A during one scene might appear in location B during the immediately following scene. Although it wasn't shown onscreen, we can infer that the character traveled between locations A and B sometime between the time intervals separating the scenes...”

In other cases the precise timing was less obvious. For example, when storylines were interleaved in the narrative (as in our replication study’s stimulus), or when a long chronological gap was implied between two scenes, we attempted to use our best judgment.

With respect to “confidence” in our annotations, we of course concede that (despite our best efforts) our annotations may have errors, and that there may be some ambiguities particularly with regard to the timing of offscreen events (which, by definition, were never depicted in the stimuli). We have attempted to address this concern in several ways:

- We have published all of our annotations for both the main experiment and replication experiment (<https://github.com/ContextLab/prediction-retrodiction-paper>).
- Any analyses of the *numbers* (or relative numbers) of past and future events, relative to the just-watched segment, are robust to the specific lags assigned to offscreen events (or other events), as long as we have estimated the event timings accurately with respect to the segments participants watched.
- Since there is no “ground truth” number of offscreen events, this presents another challenge for considering the relative numbers of “possible” events participants “could have” responded with. This makes it impossible to estimate any “hit rates” (i.e., the proportions of events that appeared in participants’ responses divided by the total number of events) that involve offscreen events. Instead, we focused our analyses on the absolute (i.e., unnormalized) *numbers* of events (page 13)
- We have added a section reporting an inter-rater reliability analysis (pages 35–36) that includes a discussion (with examples) of how ambiguities were resolved prior to carrying out our analyses.

Our overall approach to generating annotations of the stimuli that were not always specifically or objectively defined was similar to prior work with these sorts of stimuli (e.g., Chen et al., 2016; Baldassano et al., 2017; Heusser et al., 2021; and others).

5. I did not find that Figure 5, Panel E was explained very clearly. Perhaps more importantly, though, it is quite difficult to see the key data (i.e., the colored sliver at the far left of the gray bar) in these plots. I am not really sure what to recommend to the authors that would “improve” the plot, but it really is not a great viewing experience for the reader.

We appreciate the reviewer’s concern. We agree that the presentation style is quite dense, although we also (like the reviewer) struggled to come up with viable alternatives. We have updated the Figure 5 caption with some clarifications for the reader regarding how to read those plots:

“Intuitively, the widths of the rectangles at each lag denote the total number of events at each possible lag. The darker shading denotes the proportions of events that participants retrodicted or predicted, and the lighter shading denotes the proportions of events that participants “missed” in their responses.”

6. I am assuming that Figure 8 is comprised of data from the prior analyses just collated into a plot, but this isn’t clear. If it is hypothetical, this should be specified.

We have added a note to the figure’s caption (now Figure 9) to clarify that the data shown in the figure are hypothetical.

7. A larger issue I have with these data is that I am led to question the generalizability of the bias toward retrodiction due to the stimulus itself. One of the major analyses in this paper shows that the stimulus itself (i.e., references made by the characters) directs the viewer more toward the past than toward the future (Fig. 5). Further, these references seem to directly influence participants’ behavior in the experiment (Fig. 6). How, then, can we be at all confident that the “psychological arrow of time” directs us toward the past in a general sense, and not simply as a matter of cueing in this particular experiment? The results of this study seem pretty unsurprising given the aforementioned bias toward the stimulus itself referencing the past, which leads me to wonder why we would expect any other result here. At the very least, I think that this is a major enough complication with generalizability that the broader statements made throughout the paper such as “an underlying knowledge asymmetry in favor of the past” should perhaps be dialed back a bit. To be fair, the authors lay out a nice logical story for why past knowledge should exceed future knowledge in the introduction. However, there are in my view pretty major complications with the stimulus itself that limit one’s ability to make strong statements on the back of the data, and perhaps even lead to questions about the impact or importance of the findings in the absence of a more temporally-agnostic stimulus.

The core point the reviewer is raising is similar to one that several other reviewers raised: to what extent are the biases in one particular television episode reflective of more general biases (or lack thereof) in everyday life?

To help answer this question, we first ran a pre-registered replication study (using a new stimulus and participant pool). We show that the characters in the new stimulus *also* show a bias in favor of the past (which also tracks with participants' behaviors, as in our original study). We also ran a large meta analysis of a variety of texts, ranging from transcripts from television shows and movies, to novels, to spoken and written natural conversations. We collectively analyzed millions of documents. We found that, across the variety of datasets and sources we examined, references to past events are significantly more common than references to future events. We conclude that this temporal asymmetry in conversations may be a fundamental characteristic of human communication. To the extent that human communication is a fundamental feature of our real-world experiences, we expect that our experiences should also show asymmetries (beyond our own psychological arrow of time).

With respect to whether our results are "surprising" or "unexpected," that's of course difficult to determine objectively. However, we draw the reviewer's attention to several points:

- Based on prior work on statistical learning of sequences, we would have expected retrodictions and predictions of more naturalistic stimuli to be roughly symmetric. From this perspective, our results are at the very least not what *we* expected!
- Prior work using temporally "symmetric" sequences (i.e., sequences for which observing part of the sequence tells the observer just as much about prior and future states) would seem to imply that we should be equally good at retrodicting the unobserved past versus predicting the unobserved future. Our study suggests that this is *not* the case— people seem to be better at retrodicting than predicting unobserved events in the more complex and "naturalistic" stimuli we used in our experiments.
- One could imagine that television shows (and other media including films, works of fiction, etc.) could often be designed to be *entertaining* as opposed to specifically attempting to be *realistic*. We see this as a limitation of our prior manuscript, which (as the reviewer correctly notes) might conceivably not have generalized to other stimuli or been reflective of real-world behaviors or phenomena. We think that the pre-registered replication experiment in our revised manuscript helps to strengthen our case that our results were not a "fluke" driven by some idiosyncratic property of the specific stimulus or participants from our original experiment. Further, the meta analysis we carried out on tens of millions of conversations from a diverse range of fictional and real-world texts helps to strengthen our case that temporal asymmetries

are substantially more widespread than the one television show episode we examined in our prior manuscript.

8. Although the discussion is certainly the place for speculation, it felt very speculative. In particular, there were mentions of the “‘thermodynamic’ arrow of time” and “detective and forensic science” which, while interesting, are not as relevant as psychological and neuroscientific phenomena that could be discussed here. In line with my first comment about the introduction, this section was an interesting read, but it unfortunately felt shallow in terms of relating to and citing relevant literature.

First, to help “deepen” our discussion, we also have added some review of how people think about and communicate about the past and future (pages 25-26), and of how people perceive time, and how we move through time in our memories of past events or imaginings of future events (pages 27–28). To clarify how we have organized our discussion section more broadly, we can also provide some additional explanation and background here.

The “psychological arrow of time” refers to the notion that, in the present moment, we know more about our *own* past than our future, since the memories we have in the present tell us about our past but not about our future. Our work shows how our psychological arrows of time can essentially be “communicated” to other people. This appears to drive our main finding that participants are better at retrodicting the unknown (unobserved) past than they are at predicting the unknown future.

In our discussion section we tried to situate our main findings within the context of several relevant literatures. First, how do we “transmit” our knowledge and memories to other people, in the general sense? And if our own knowledge and memories are “biased” (e.g., because we know more about our own past than our own future), does this bias show up in what or how people communicate? We tie in a number of studies that get at how knowledge and memories are transmitted across people (page 26).

Next, how “universal” are these temporal biases? We discuss how typical studies of sequence learning (including sequence retrodiction and prediction) have relied on lab-made stimuli that are temporally symmetric, such as random sequences, first-order Markov processes, or other stationary time series. Behaviorally, people do *not* show temporal asymmetries in retrodicting or predicting these types of sequences. So what is different about the “sequences” used in our study? Or are the temporal biases we observed simply a “fluke” that is not actually reflective of the real world or of everyday life?

The question of why there might be temporal asymmetries in more “naturalistic” sequences (narratives, real-world experiences, etc.) led us down a rabbit hole of work spanning several fields outside of psychology, from philosophy, to pure math, to statistical physics, and beyond. These literatures are complex, diverse, and vast. We attempted to distill down the most relevant ideas from our literature search. A core set of ideas relates to why the universe itself may have temporal asymmetries, which could in turn affect how we perceive time and/or how our memory systems evolved. Even though most laws of physics are time-symmetric, the second law of thermodynamics (essentially: the total entropy in a closed system cannot decrease) is *not* time-symmetric. An implication of a universe with non-decreasing entropy is that the past will tend to be simpler than the future (this is what the “thermodynamic arrow of time” concept is getting at). While we agree that it is a bit untraditional to reference such different literatures in what is ostensibly a psychology paper, we also think that work makes a deep and important point about where temporal asymmetries in our knowledge might *come* from. We also agree that these ideas are speculative, and we’ve tried to word the discussion to clearly indicate when we were speculating versus referencing specific findings from our study or prior work.

In our concluding paragraph of the discussion section we suggest that reconstructing the unknown past goes beyond an individual person’s own experiences, but rather it appears to be a major focus of many areas of human inquiry (including not just detective and forensic science, but also history, anthropology, geology, among others). Our goal there is to draw connections with other aspects of the human experience (e.g., *why* humans devote so much effort to reconstructing the unobserved past) as opposed to leveraging specific insights from those fields.

9. A very minor nitpick, but it would be nice to include a quick rundown of odds ratios, at least in the methods section. This is certainly an appropriate way of analyzing these data, but I find it likely that a number of prospective readers might not immediately grasp what was being evaluated here.

We have added a paragraph to our methods section (page 40) describing how we computed the odds ratios and how they should be interpreted; here is an excerpt of the odds ratio “rundown” that we added:

“...the odds ratios reflect the odds (calculated as $p/(1-p)$, while p is the probability that the outcome occurs) of a particular outcome (e.g., making a response about a particular event) given a scenario (e.g., the event occurred *prior* to the just-watched segment) compared with the odds of the outcome occurring in the alternative scenario (e.g., the event occurred *after* the just-watched segment).”

Reviewer #3 (Remarks to the Author):

Since the time of Ebbinghaus, memory research has largely focused on carefully controlled lists of stimuli. One of the important phenomena this work has revealed is the contiguity effect, the finding that, all else equal, memory for an event brings to mind memory for nearby stimuli. Over roughly the last decade or so, there has been a movement to study human memory under more naturalistic circumstances. Much of this work has relied on participants experiencing narratives, such as radio shows or oral story-telling or, as in this study, television programs. Because the study materials are so different, it is difficult to assess the source of differences between laboratory memory studies and studies using narrative material. After all, narratives often have a rich structure with layers of semantic meanings and emotional and cultural shadings. It is difficult to measure these variables from narratives and thus difficult to understand their separate and cooperative effects. The present study presents an ingenious method to estimate the connectedness of the narrative. One might say that this study measures the contiguity effect in the complete absence of memory for the study materials.

Interesting point!

Participants are presented with brief clips from an unfamiliar television program and asked to predict events that will follow the clip and retrodict events that preceded the clip. In this way it is possible to trace out something analogous to a contiguity effect that is solely attributable to the participants' expectations about the statistical structure of the narrative. The primary result is that, whereas the contiguity effect in laboratory memory experiments is typically asymmetric favoring memory in the forward direction, the ability to predict unobserved events from a narrative is asymmetric in the backward direction. On its face, this result is not only inconsistent with the contiguity effect from laboratory memory experiments, but also from laboratory experiments that directly evaluate participants' abilities to estimate the past and the future from Markov sequences of stimuli. Jones & Pashler (2007) observed no evidence for an asymmetry, in either direction, when participants were asked to judge the stimulus that followed or preceded a cue stimulus.

We have added some discussion of this point (pages 27–28). Of note, we view the two sets of findings as not in conflict, but rather reflecting different aspects of memory:

“For example, a well-studied phenomenon in the episodic memory literature concerns how remembering a given event cues our memories of other events that we experienced nearby in time (i.e., the contiguity effect; Kahana, 1996). Across a large number of studies there appears to be a nearly universal tendency for people to move forwards in time in their memories, whereby recalling an “event” (e.g., a word on a

previously studied list) is about twice as likely to be followed by recalling the event that immediately followed as compared with the event immediately preceding the just-recalled event (Healey and Kahana, 2014). Superficially our current study appears to report the opposite pattern, whereby participants display a backwards temporal bias. However, the two sets of findings may be reconciled when one considers the frame of reference (and current mental context; e.g., Howard and Kahana, 2002) of the participant at the moment they make their response. In our study, participants observe an event in the present, and they make guesses about what happened in the unobserved past or future, relative to the just-observed event. (Our findings imply that participants are more facile at moving backwards in time than forwards in time, relative to “now.”) In contrast, the classic contiguity effect in episodic memory studies refers to how people move through time relative to a just remembered event. The forward asymmetry in the contiguity effect follows from the notion that the moment of remembering has greater contextual overlap with events after the remembered event from the past, including the moment of remembering, than events that happened before it (for review also see Manning et al., 2015; Manning, 2020). In other words, our current frame of reference appears to exhibit a sort of “pull” on our thoughts, such that thoughts about recent experiences still lingering in our minds drag us towards the recent past, but after thinking about the more distant past we are dragged (relatively) forward in time back to “now.””

So what accounts for the counterintuitive result in the present study. The authors argue that the fact that in the narrative, characters talk about past events more frequently than future events. This at least partially accounts for the backward asymmetry, which seems to at least be reduced for events that were not referenced. Moreover, the authors argue that references enhance the ability to predict events surrounding the referenced events. For instance, if a character at work mentions that they shoveled their driveway that morning, one might guess that it snowed in the night and that they drove to work.

Exactly– because the *characters* know their own pasts (better than their not-yet-experienced futures), their conversations reflect this asymmetry in their own knowledge. In turn, our participants pick up on those asymmetries. Essentially, the characters’ temporal biases (in their knowledge about their own lives) are being transmitted to the participants as they watch the television show.

In summary, this paper is methodologically innovative and tackles a problem that is of importance in the study of human memory. The analyses are careful and sound. Like many studies that use cultural artifacts as study materials, it is unclear how well this generalizes to actual real world experiences, or even other cultural artifacts. Narratives are crafted to be unpredictable; a television show that is too

predictable is boring. However, for the narrative in this study at least, it seems that asymmetry in the ability to infer events in time was at least to some extent attributable to the memory conveyed by the characters in the program.

The question of generalizability is an important one. As the reviewer suggests, is there something special about the particular television show we selected for our study? Or about television, or media crafted to be entertaining, in general (e.g., whereby temporal asymmetries may increase the audience's enjoyment by making the narratives less predictable)? We addressed this question in two ways in our revision.

First, we wanted to verify that our core experimental findings generalized beyond the specific stimulus (and group of participants) in our original experiment. We ran a pre-registered replication experiment (using a new stimulus and a new group of participants). Our replication experiment showed that (a) characters in the new stimulus also show temporal biases towards the past in their conversations, and (b) participants show a bias towards retrodiction (versus prediction) that is guided by the characters' conversations. This increases our confidence in the core findings from our original experiment (what we refer to as our "main experiment" in our revised manuscript).

Second, we sought to test whether temporal biases in conversations are a general property of conversation, or whether it instead might be specific to television shows, media or narratives intended to be entertaining, etc. To this end, we carried out a meta analysis of tens of millions of conversations. The datasets we analyzed were drawn from "entertaining" media like transcripts of television shows, movies, and novels, as well as "natural" spoken and written conversations. We found that, across these very different types of conversations, people (real and fictional) refer to events in the past 1.45 times more often than they refer to events in the future. Of course this does not mean that every *individual* conversation has these same biases. For example, one of the datasets in our meta analysis comprised transcripts of conversations between people discussing future plans. Those conversations showed a temporal bias in favor of *future* references. Similarly, individual conversations in each dataset showed a range of temporal biases, whereby some were more about the past, some were more about the future, some were evenly balanced, and others were solely about the present. Nonetheless, our meta analysis suggests that, overall, people tend to talk more about the past than the future. In other words, this bias towards the past seems to be a general tendency in human conversation as opposed to merely a "fluke" in the particular stimuli we happened to choose for our experiments.

One might ask if temporal inference of events is asymmetric for narratives that do not explicitly convey information about the past or the future (assuming the Jones & Pashler results are the last word on genuinely Markov processes). Even those without characters describing past events. Simple statistical structures may yield asymmetric predictions from a normative model. For instance, suppose that a stimulus Y happens with relatively high probability according to a Poisson process. The stimulus train is assembled such that every once in a while, Y is preceded by X at precisely 1 second. That is $P(Y \text{ presented } 1 \text{ s in the future} \mid X \text{ presented now}) = 1$, whereas $P(X \text{ was presented } 1 \text{ s in the past} \mid Y \text{ presented now})$ is very small. Similarly, taking into account the imprecision of temporal memory predictions that can be formed from the sequence

$X Y Z$

are asymmetric. Starting from Y , the time of X is more uncertain than the time of Z , simply because of our ability to resolve temporal intervals.

We agree that it is (potentially) possible to construct sequences, along the lines of what the reviewer is proposing, that have inherent temporal biases (e.g., whereby either the past or future is easier to predict because of the rules of how the sequences are formed). We think this is part of why these questions are interesting!

As the reviewer notes, (first order) Markov processes (and other stationary processes) are symmetric in time: i.e., given an observation partway through the sequence, we have equal information about the sequence's past and future states (Cover, 1994). Prior work (e.g., the Jones and Pashler study mentioned by the reviewer) suggests that there are no temporal biases in people's inferences for these sorts of sequences.

The conflict is that in people's everyday experiences, people *do* show biases. Part of this is due to the "psychological arrow of time" concept we bring up in the introduction: in the present moment we have memories of our past, but not of our future, and so we are more equipped to infer what happened in our own pasts than what *will* happen in our own futures. The biases people show for real-world experiences (and in our experiments) also seem to arise due to properties of their *experiences* (e.g., whereby experience itself gives us more "information" about the past than the future, conditioned on the present). An interesting challenge is to try to tease apart the respective contributions of psychological processes (internal to our minds and brains) versus external processes or properties of people's experiences (or the experimental stimuli).

If we had simply asked participants to "guess" what they had experienced in their past versus what they might experience in the future, the experiment would have been trivial—people could simply draw on their memories to recall what they had done in the past, but

they would have no particular or specific way of predicting their future experiences other than drawing on general schemas they had learned. (E.g., we would expect that unlikely or low probability events, such as the experimenter suddenly interrupting the testing session and breaking out in song, would be highly unlikely to show up in participants' predictions of the future, but they'd almost certainly show up in their recollections of the recent past!)

Instead, we designed our task such that participants could *not* rely on their own psychological arrow of time to infer the past or future. Therefore any biases in participants' inferences had to either come from some property of the stimulus itself, or perhaps some biases inherent to our memory systems in general that are independent of *what* we experience. We found that participants' biases towards the past seemed to come at least in part from the stimulus itself.

Whether "real world events" are more like symmetric Markov sequences or more like event sequences in narratives is a deep question. We think the question actually touches on fundamental properties of our universe, such as the second law of thermodynamics. Essentially, the second law of thermodynamics (i.e., total entropy does not decrease in a closed system) says that the past is fundamentally more "predictable" than the future. In other words, the "toy" example the reviewer suggests, whereby either the past or the future is inherently more reliable, is what our *actual* universe is like: the past (of our universe) is fundamentally more constrained (and therefore "easier" to infer) than the future of our universe. An in-depth discussion of this point goes beyond the scope of our manuscript, but it's what we mean by the "thermodynamic arrow of time" in our discussion section (pages 26–27). In follow-up work we are using simulations to explore potential links between the thermodynamic and psychological arrows of time.

As an aside, the reviewer may be interested to know that even the *seemingly* asymmetric example sequence they proposed is actually *not* asymmetric! When we say that a sequence is asymmetric, we mean that, conditioned on observing one part of the sequence, it's easier to infer either the past than the future parts of the sequence (for backwards-asymmetric sequences), or it's easier to infer the future than the past (for forwards-asymmetric sequences). But that conditioning aspect is defined over *all possible* parts of the sequence—not just isolated examples. In the example the reviewer proposes, computing the "symmetry" of the sequence would require averaging over some time points where X is present, some where Y is present, and some where neither are present. If restricted to just the timepoints when Y was present, the reviewer's intuition is correct that it would be easier to retrodict that X occurred 1 second in the past, than to predict what might happen 1 second in the future. But for exactly the same number of timepoints (where X is followed by Y), at the

X timepoints it will be easier to *predict* that Y occurred 1 second in the *future* than to retrodict what might have happened 1 second in the past. So when we average over all timepoints in the reviewer's example sequence, those two cases perfectly cancel each other out and we see that the sequence is symmetric.

Constructing *truly* asymmetric sequences is actually a surprisingly difficult endeavor. One approach is to construct non-stationary sequences— e.g., where the underlying “rules” change systematically over time. This is more along the lines of how we have come to think that real-world experiences might “work.” Specifically, because entropy in our universe is non-decreasing on average, the past tends to be more constrained than the future (conditioned on the present, and averaging over all possible “nows”— e.g., it is still possible to have “local” examples where the future is more constrained than the past). Further, since the average total entropy in the near past *always* tends to be lower than total entropy in the near future (no matter which time we're considering), the “averaging over all possible nows” step *doesn't* cancel out the local asymmetries, and local violations are always overshadowed by other instances that conform to the modal pattern.

Reviewer #4 (Remarks to the Author):

The authors report evidence that people are better able to retrodict other people's pasts than predict other people's future. Past work has shown similar effects for people's own personal experience, but this is the first work to show that the effects extend to how we understand other people and their experiences. The authors further showed that the temporal asymmetry in character retrodictions/predictions was related to (i) characters' tendencies to refer to past events more than future events in their ongoing conversations, and (ii) associations between temporally proximal events. There is a lot to like about this paper: the results are compelling, the experimental design is creative, and it is a novel finding. There are a few concerns that if addressed in a revision would strengthen the manuscript and make it better suited for publication.

We thank the reviewer for the positive feedback!

1. Television writers presumably spend a good chunk of time trying to write scripts that viewers cannot easily predict. When as a viewer you can easily predict what happens next in a show, it's less entertaining—people frequently complain about shows that are “too predictable.” Simultaneously, tv writers want to make the story easy to remember so people can jump in and out, or remember what happened in the last episode so they can enjoy the next episode a week later. This all makes me wonder if the temporal asymmetries observed here generalize to other scenarios that are not scripted and not-pre-designed to be more or less predictable vs. retrodictable. If the authors were able to replicate their

results with footage of people in unscripted, everyday conversations, that would very compelling. Or at the very least, this would be good to address as a limitation in the discussion section.

This is an important point, and one also raised by Reviewer 3. Fundamentally, we see this as a question about how “representative” the temporal bias towards the past is of stimuli (and/or real-world experiences) in general, versus whether it is simply a property of the particular stimulus we happened to choose, or of stimuli that are specifically created to be “unpredictable” for entertainment purposes. For conveniences, we’ve copied our response below:

The question of generalizability is an important one. As the reviewer suggests, is there something special about the particular television show we selected for our study? Or about television, or media crafted to be entertaining, in general (e.g., whereby temporal asymmetries may increase the audience’s enjoyment by making the narratives less predictable)? We addressed this question in two ways in our revision.

First, we wanted to verify that our core experimental findings generalized beyond the specific stimulus (and group of participants) in our original experiment. We ran a pre-registered replication experiment (using a new stimulus and a new group of participants). Our replication experiment showed that (a) characters in the new stimulus also show temporal biases towards the past in their conversations, and (b) participants show a bias towards retrodiction (versus prediction) that is guided by the characters’ conversations. This increases our confidence in the core findings from our original experiment (what we refer to as our “main experiment” in our revised manuscript).

Second, we sought to test whether temporal biases in conversations are a general property of conversation, or whether it might be specific to television shows, media or narratives intended to be entertaining, etc. To this end, we carried out a meta analysis of tens of millions of conversations. The datasets we analyzed were drawn from “entertaining” media like transcripts of television shows, movies, and novels, as well as “natural” spoken and written conversations. We found that, across these very different types of conversations, people (real and fictional) refer to events in the past 1.45 times more often than they refer to events in the future. Of course this does not mean that every *individual* conversation has the same biases. For example, one of the datasets in our meta analysis comprised transcripts of conversations between people discussing future plans. Those conversations showed a temporal bias in favor of

future references. Similarly, individual conversations in each dataset showed a range of temporal biases, whereby some were more about the past, some were more about the future, some were evenly balanced, and others were solely about the present. Nonetheless, our meta analysis suggests that, overall, people tend to talk more about the past than the future. In other words, this bias towards the past seems to be a general tendency in human conversation as opposed to merely a “fluke” in the particular stimuli we happened to choose for our experiments.

The reviewer’s suggestion to carry out an additional experiment involving natural conversations between participants is very interesting. We struggled, though, to come up with a viable experiment. For example, suppose we had pairs of participants engage in conversation, and then periodically paused the conversations to (separately) ask the participants to retrodict or predict what had happened in the other participant’s past or future? Or, suppose we had a third participant listen to snippets of another pair’s conversations, and then retrodict or predict past or future parts of the conversation? The main challenges we saw are that (a) it would be difficult to actually validate those responses (e.g., we would need to collect some sort of information about the entirety of each participant’s past *and* future experiences throughout their entire lives), and (b) the scope of potential conversation topics is so broad that it’s difficult to imagine what might serve to constrain participants’ responses. Further, whereas television shows and movies tend to compress the “interesting” aspects of experiences (often skipping over mundane details), interesting events in real-life experiences tend to be relatively infrequent. Given these challenges, we instead chose to (a) replicate our original experiment using a new (similar) stimulus, and (b) carry out meta analyses on a range of real and fictional conversations, including “natural” (unscripted) conversations. We focused in our meta analyses on identifying *references* to past versus future events, since our experiments indicated that those references are a primary driver of people’s retrodictions and predictions.

2. Is it possible to dive even deeper into the features of the story that are better retrodicted than predicted? Is it interpersonal events that are best retrodicted? Characters’ internal thoughts/feelings/beliefs? Key narrative transition points? Something else? The authors note that it is partly what characters say about the past, but it would be interesting if the effect is driven by certain content communicated.

This is an interesting question. To begin to explore this idea, we manually labeled events in the main experiments as social or non-social. We assigned these labels according to whether each event involved multiple people interacting (e.g., talking or doing an activity together) or not. As in the reference effect analysis in our main text, we ran mixed effect models on the hit rates, controlled for lags, on the onscreen events only. We found no significant interactions

between social/non-social and retrodiction/prediction (OR = 1.20, Z = 0.78, $p = 0.44$). Next we tested whether social or non-social events were communicated more than the other by counting the numbers of social vs non-social events in character past and future references (Table 1). Again we found no interaction between social/non-social and past/future references (main experiment: $\chi^2(1) = 0.20, p = 0.66$; replication experiment: $\chi^2(1) = 2.30, p = 0.13$).

Table 1: Counts of past/future references of social/non-social events in both main and replication experiments' stimuli

		non-social	social
main	past	30	22
	future	15	15
replication	past	17	29
	future	0	7

We also considered a variety of other approaches. In one of our analyses, we used text embedding models to (formally) characterize the “content” of each event or scene in the television episodes. In principle one could imagine carrying out some sort of recursive feature elimination analysis (e.g., removing one feature at a time, and seeing how the correlations between the text embeddings for the actual episode events vs. participants' responses changed). However, in practice, this sort of approach is fundamentally under-specified for our setup. For example, suppose that we could identify three features whose values were monotonically increasing (for a particular event) in the embeddings for both the original stimulus event and the average retrodictions of that event. This would yield a strong positive correlation (seemingly an excellent match!). But whether those features are actually meaningful, and *what* those features mean, really depends on what the other feature values are. If those three features had values very near zero (but just happened to have the same ordering for a particular event), focusing on those features would yield an artificially high correlation. In other words, more generally, naive approaches like finding the subset of features that display the maximum effect, or maximize a similarity metric, still could not tell us whether those features were actually *meaningful* with respect to driving (or helping to explain or account for) participants' behaviors.

We also considered the reviewer's idea about examining narrative transition points. In some of our prior work (e.g., first developed in Heusser et al., 2021, *Nature Human Behavior*), we have used text embeddings of each sliding window of the narrative to characterize its "content," and then we apply hidden Markov models to automatically segment the content into discrete events. We also considered manual segmentation approaches. Unfortunately this line of analysis did not seem well-suited to our current study, for several reasons. For example, in preparing the stimulus for presentation to the participants we had *already* (manually) divided it into segments based on narrative transitions and scene boundaries. While we did find distinct sub-segment events, we felt (from manual examination) that these sub-segment events did not reflect major narrative transitions— e.g., they seemed to be far overshadowed by the segment-level transitions we had already divided the story into for presentation. For these reasons, we felt we would lack appropriate power in our analysis: there are only around a dozen narrative transitions in each stimulus we used in our experiments, and all of them were confounded with the segment boundaries that determined how we showed the narratives to the participants. In sum, although we think the idea of exploring the effects on inference of narrative transition points is very interesting (and a potentially promising direction for future work!), we feel that our current study cannot provide meaningful insights into that particular aspect of these phenomena.

Another approach to explore these ideas is to examine participants' responses closely by hand. We could then attempt to see if any patterns emerged. Essentially this is what we ended up doing when we wrote our initial paper, as well as in our replication experiment. Nearly all of the responses participants made refer to events involving some sort of interpersonal interaction between the characters. From our examinations we didn't notice any particular "types" of interactions that participants tended to respond with more often than others. We did notice that participants almost never predict or retrodict the content of the conversations characters had. In other words, even though characters frequently refer to past or future events in their conversations, our participants never predict that "character [X] is going to talk about this later." We quantified this in our "referenced versus referring events" analysis reported in Figures 7 and S11.

3. A replication would also allow the authors to potentially test some questions about the psychological processes driving the effect. In social cognition research, a common finding is that people "use the self" to simulate other people (see some relevant citations below). And, they use this strategy more for people they believe are similar to themselves. Given that your findings parallel how people's retrodictions/predictions about themselves work, it seems possible that this "simulation from self" strategy is at work here (particularly if analyses related to the question above suggest retrodiction/prediction for "invisible mental states" correspond with this temporal asymmetry). If a large online study was run (either with the same

stimuli reported here or different stimuli that's unscripted), you could also ask participants how similar they feel to each character and test whether that modulates the results. Alternatively, if the authors do not want to run such a study, the simulation from self strategy could be proposed or discussed in the discussion.

Tamir, D. I., & Mitchell, J. P. (2013). Anchoring and adjustment during social inferences. *Journal of Experimental Psychology: General*, 142(1), 151.

Meyer, M. L., Zhao, Z., & Tamir, D. I. (2019). Simulating other people changes the self. *Journal of Experimental Psychology: General*, 148(11), 1898.

We carried out two additional experiments to help further test our core claims. The first new experiment was a replication of our original experiment aimed at further testing our initial findings. Our second new experiment was a meta analysis aimed at examining how “representative” the temporal asymmetries we observed in our original stimulus were of different fictional and real conversations. To reign in the scope of our current study, we chose not to run the additional large *n* study the reviewer suggested above, although we agree that such a study could be interesting (e.g., in followup work).

It's interesting to speculate about what the psychological processes underlying our results might be. We suspect that the reviewer's intuition is correct— e.g., if participants “identified more” with particular characters, they might “trust” those characters' references more, or perhaps might more accurately form inferences about unobserved events in those characters' lives. Similarly, if participants were especially familiar with the “schema” reflected in the narratives (e.g., in our replication study using *The Chair*, participants might be familiar with the tenure process or aspects of academic life depicted in the story), they might be able to form more accurate inferences that drew on their domain knowledge. That said, this line of reasoning felt somewhat “subjective” to us, as neither our study nor the one the reviewer is proposing seems to *directly* measure psychological processes per se; rather, these experiments seem to measure participants *behaviors* (which we think *indirectly reflect* psychological processes), but our best hope is to speculate about what those processes might be, as opposed to objectively measuring them. In any case, at the reviewer's suggestion, we've added some discussion of the reviewer's proposal, along with citations of the suggested references (page 26):

“Some recent work (e.g., Tamir & Mitchell, 2013; Meyer et al., 2019) also suggests that people might use “mental simulations” of how other people might respond in

particular situations (e.g., in the future), or of which sorts of prior experiences might have led someone to behave a particular way in the present.”

REVIEWER COMMENTS

Reviewer #2 (Remarks to the Author):

I must say that the authors have gone above and beyond what I would consider to be adequate in responding to my comments, as well as other reviewer comments. The replication experiment and improvements throughout the manuscript have more than cleared the bar for me to be satisfied with the revision. The authors were both thorough and thoughtful in their responses and edits. I have no further comments or concerns.

Reviewer #3 (Remarks to the Author):

In my view the changes to the revision---including the new experimental work and the meta-analysis---have strengthened the original submission.

I recommend publication.

Reviewer #4 (Remarks to the Author):

The authors have completed a very thorough response to the concerns raised. It's exciting to see the new results from the new meta-analysis. Congratulations on a very interesting paper.

Reviewer #5 (Remarks to the Author):

In reviewing the text analysis parts of the manuscript, the authors used word embeddings to compare similarities of participant retrodictions and predictions as well as other methods to calculate relative focus on past versus future time in several corpora. The methods are definitely interesting and creative, but the manuscript lacks sufficient detail and justification on the methods.

First, for the word embeddings in the main and replication studies, the calculations of 'precision' and 'convergence' are unconvincing. Based on my reading, it would seem that the only difference between the two is that one uses the median similarity and the other uses the mean similarity which wouldn't create two distinct metrics. Clarity and justification for the calculations are needed.

Second, for the meta-analysis, there really isn't information about it to properly judge it. Context/type of corpus isn't considered which would be an important consideration given that some topics would naturally skew to past or future talk. The analysis isn't explained fully so the effect sizes don't necessarily make sense. It seems like Odds Ratios are calculated but that is unclear. One of the effect sizes is reported as effect size: 0.68 ± 0.10 ; CI: 0.59 to 0.65, which is statistically impossible. Finally, the method for identifying past versus future is overly complex as existing reliable methods include LIWC or dependency parsing (udpipe package in R).

Reviewer #2 (Remarks to the Author):

I must say that the authors have gone above and beyond what I would consider to be adequate in responding to my comments, as well as other reviewer comments. The replication experiment and improvements throughout the manuscript have more than cleared the bar for me to be satisfied with the revision. The authors were both thorough and thoughtful in their responses and edits. I have no further comments or concerns.

We appreciate the reviewer's positive assessment of the work.

Reviewer #3 (Remarks to the Author):

In my view the changes to the revision---including the new experimental work and the meta-analysis---have strengthened the original submission.

I recommend publication.

We appreciate the reviewer's positive assessment of the work.

Reviewer #4 (Remarks to the Author):

The authors have completed a very thorough response to the concerns raised. It's exciting to see the new results from the new meta-analysis. Congratulations on a very interesting paper.

We appreciate the reviewer's positive assessment of the work. As recommended by the editorial team, we now refer to the "meta-analysis" as a "large-scale analysis" in our revised manuscript.

Reviewer #5 (Remarks to the Author):

In reviewing the text analysis parts of the manuscript, the authors used word embeddings to compare similarities of participant retrodictions and predictions as well as other methods to calculate relative focus on past versus future time in several corpora. The methods are definitely interesting and creative, but the manuscript lacks sufficient detail and justification on the methods.

First, for the word embeddings in the main and replication studies, the calculations of 'precision' and 'convergence' are unconvincing. Based on my reading, it would seem that the only difference between the

two is that one uses the median similarity and the other uses the mean similarity which wouldn't create two distinct metrics. Clarity and justification for the calculations are needed.

A major distinction between our precision and convergence measures is that they are calculated using different data. They were also designed to get at different conceptual aspects of the data. We define the two measures and provide some conceptual intuitions about each on pages 8–10 of our revised manuscript (as well as below, for convenience). We have also updated the descriptions of each measure in Figure 3A to help clarify what each measure is capturing.

Formally, we defined both measures as follows:

- Precision is defined as the median cosine similarities between the embeddings of (a) a participant's retrodiction or prediction response for the target segment and (b) other participants' *recalls* of that same target segment. Fundamentally, this is a comparison between *inferences* (about the unobserved past or future) and *recalls* (of just-watched content).
- Convergence is defined as the mean cosine similarity between the embeddings of a participant's responses to a target segment and other participants' responses of *the same type* to the same segment. Fundamentally, this is a comparison between the same types of responses (i.e., retrodictions *or* predictions *or* recalls) across participants.

Conceptually, the two measures are designed to characterize different aspects of participants' responses:

- By measuring the similarity between participants' inferences about past/future segments and the ways other participants *recalled* the same segment, *precision* is designed to measure the extent to which retrodictions and predictions captured the conceptual content that *other* participants remembered about the same segments. In other words, we're trying to capture the extent to which a given participant is retrodicting or predicting the content from a segment that people tend to remember about that segment after just having watched it. (We exclude the retrodicting/predicting participant from the comparison pool to avoid circularities in the analysis.)
- By measuring similarities across participants for the same response type, to the same segment, *convergence* is designed to measure agreement across individuals. In other words, we want to capture (for a given segment and response type) the extent to which different people responded in a similar way.

Second, for the meta-analysis, there really isn't information about it to properly judge it. Context/type of corpus isn't considered which would be an important consideration given that some topics would naturally skew to past or future talk. The analysis isn't explained fully so the effect sizes don't necessarily make sense. It seems like Odds Ratios are calculated but that is unclear. One of the effect sizes is reported as effect size: 0.68 ± 0.10 ; CI: 0.59 to 0.65, which is statistically impossible. Finally, the method for identifying past versus future is overly complex as existing reliable methods include LIWC or dependency parsing (udpipe package in R).

First, we acknowledge that context/type of corpus is an important consideration. We have added a note to page 22 noting that we chose the datasets partly for convenience, and partly to cover a wide range of different types of conversations. Following Serban et al. (2015), in our revised manuscript we have labeled each dataset as comprising either *constrained*, *scripted*, and/or *spontaneous* human-human conversations. Due to the limited number of datasets we used, we were unable to draw conclusions about which type(s) of conversations might be more skewed towards past or future references. Instead, we used a (more traditional) meta-analysis-like random effects modeling approach to find the general trend in those datasets.

Second, as requested, we have also added additional detail about the analysis (pages 22-24 and 41-43). Included in our revised descriptions are explanations of how we computed effect sizes (page 24).

Third, we also appreciate the reviewer's catch about the "impossible" effect size (this appears to have been a typo). In reviewing our general approach, adding the requested details, and in considering the editorial team's comment about our analysis not being a "true" meta analysis, we decided to revamp the analysis to more closely align with more traditional meta analyses:

- For each dataset, we calculated an effect size as the Past/Future ratio, defined as the proportion of references to past events (in all sentences) divided by the proportion of references to future events (commonly known as *risk ratio* or *relative risk*). Since the denominators (the total number of sentences) were the same in the two proportions, the Past/Future ratio could be simply calculated as the the number of references to past events divided by the number of references to future events.
- We now fit a random-effects model to the log-transformed ratios to characterize the overall trends across datasets.

Fourth, we appreciate the reviewer's observation that our approach to identifying past and future references is complex. Our intuition prior to carrying out that analysis was similar-

we expected that reliable tense identification was likely a “solved problem.” We experimented with a variety of other simpler or pre-packaged approaches, including LIWC and dependency parsing, as well as even *more* complicated approaches, such as asking ChatGPT to identify and count uses of past and future tense in conversations or excerpts. In practice, we found that most “off the shelf” approaches (including the ChatGPT idea we tried) performed surprisingly poorly. We assessed performance by comparing the manually identified references to past and future events from *The Chair, Episode 1* (that we used in our other analyses, e.g., of participants’ behavioral data) with automatically derived tenses using different automated approaches. In carrying out these comparisons we spot checked randomly sampled examples, attempted to construct our own “tricky edge cases,” and also examined aggregate counts across conversations within the episode. Once we verified that we could reproduce (using an automated approach) the relative numbers of past versus future references identified by hand in that one episode, we then applied the automated approach to other episodes of *The Chair*. We found that our approach appeared to generalize to those other episodes as well. Finally, after computing automated tags for the other datasets, we carried out a final set of “spot checks” on randomly excerpted utterances from each dataset to verify that the automated tags were behaving as expected. We have added a note to this effect on page 42.

REVIEWERS' COMMENTS

Reviewer #5 (Remarks to the Author):

The authors sufficiently addressed my concerns.